# A 10 m resolution land cover map of the Tibetan Plateau with detailed vegetation types

Xingyi Huang[1,#], Yuwei Yin[1,#], Luwei Feng[1], Xiaoye Tong[2], Xiaoxin Zhang[2], Jiangrong Li[3], and Feng Tian[1,4,*]

[1]Hubei Key Laboratory of Quantitative Remote Sensing of Land and Atmosphere, School of Remote Sensing and Information Engineering, Wuhan University, Wuhan, China
[2]Department of Geosciences and Natural Resource Management, University of Copenhagen, Copenhagen, Denmark
[3]Institute of Tibet Plateau Ecology, Tibet Agricultural and Animal Husbandry University, Linzhi, China
[4]Perception and Effectiveness Assessment for Carbon-neutrality Efforts, Engineering Research Center of Ministry of Education, Wuhan, China
[#] These authors contributed equally to this work.
[*] Corresponding author

**Correspondence:** Feng Tian (tian.feng@whu.edu.cn)

**Abstract.** The Tibetan Plateau (TP) hosts a variety of vegetation types ranging from broadleaved and needle-leaved forests at the lower altitudes and mesic areas to alpine grassland at the higher altitudes and xeric areas. Accurate and detailed mapping of the vegetation distribution on TP is essential for an improved understanding of climate change effects on terrestrial ecosystems. Yet, existing land cover datasets of TP are either provided at a low spatial resolution or have insufficient vegetation types to characterize certain unique TP ecosystems, such as the alpine scree. Here, we produced a 10 m resolution TP land cover map with 12 vegetation classes and 3 non-vegetation classes for the year 2022 (referred as TP_LC10-2022) by leveraging state-of-the-art remote sensing approaches including the Sentinel-1 and Sentinel-2 imagery, environmental and topographic datasets, and 4 machine learning models using Google Earth Engine platform. Our dataset TP_LC10-2022 achieved an overall classification accuracy of 86.5% with a Kappa coefficient of 0.854. By comparing with 4 existing global land cover products, TP_LC10-2022 showed significant improvements in terms of reflecting local-scale vertical variations in the southeast TP region. Moreover, we found that alpine scree occupied 13.99% of the TP region which was ignored in existing land cover datasets, and that shrublands occupied 4.63% of the TP region characterized by distinct forms of deciduous shrublands and evergreen shrublands largely determined by topography and missed in existing land cover datasets. Our dataset provides a solid foundation for further analyses which need accurate delineation of these unique vegetation types in TP. The TP_LC10-2022 and the sample dataset are freely available at https://doi.org/10.5281/zenodo.8214981 (Huang et al., 2023a) and https://doi.org/10.5281/zenodo.8227942 (Huang et al., 2023b) respectively. Additionally, the classification map can be viewed through https://cold-classifier.users.earthengine.app/view/tplc10-2022.

## 1 Introduction

The Earth's surface is physically covered by various types of land cover, including forests, grasslands, croplands, lakes, wet-
lands, etc. Accurate mapping and classification of land cover are fundamental components for Earth observations. By under-
standing the distribution and characteristics of different land cover types, land cover mapping supports the assessment of carbon
stocks, vegetation dynamics, and land-atmosphere interactions, contributing to the implementation of effective climate change
mitigation measures (Wang et al., 2022b; Liu et al., 2022; Li et al., 2018).

The advent of remote sensing technology has enabled the generation of global-scale land cover products at various resolu-
tions. For instance, products like MCD12Q1, produced using MODIS data (Friedl et al., 2010, 2002), and the ESA CCI product,
derived from sensors like MERIS (Agency, 2014), have significantly contributed to the understanding of global ecosystem re-
sponses to climate change. However, their spatial resolutions are at hundreds of meters, unable to provide an accurate represen-
tation of the land surface conditions (Tian et al., 2021), particularly in spatially heterogenous regions, such as the mountainous
southeast Tibetan Plateau (TP) (Yang et al., 2017; Grekousis et al., 2015). In response to this limitation, several medium- to
high-resolution land cover products have been created using satellite images from Landsat and Sentinel-2. Notable examples
include GlobeLand30 (Chen et al., 2021, 2015), FROM_GLC30 (Gong et al., 2013), GLC_FCS30 (Zhang et al., 2021b) based
on Landsat, and FROM_GLC10 (Chen et al., 2019), Dynamic World (Brown et al., 2022), Esri Land Cover (Karra et al., 2021),
and ESA WorldCover (Zanaga et al., 2022) based on Sentinel-2. However, these products use different classification systems,
resulting in large divergence in certain regions (Shi et al., 2023; Hua et al., 2018), and are often inadequate to reflect the diverse
and unique land cover types for important ecosystems (Liu et al., 2023a), such as those in the TP.

Renowned as the "Third Pole" of the world (Shukla and Sen, 2021), TP holds a dual significance as a sensitive area and
an indicator zone for global climate change (Hua et al., 2021; Li et al., 2022; Trew and Maclean, 2021; Pepin et al., 2022).
It hosts a variety of vegetation types, ranging from broadleaved and needle-leaved forests at the lower altitudes and mesic
areas to alpine grassland at the higher altitudes and xeric areas. However, many of the unique vegetation types in TP are not
well represented in existing land cover datasets. For example, the alpine scree ecosystem in the transitional zone from alpine
grasslands to bare rocks at very high altitudes and the shrubland ecosystem in the transitional zone from forests to grasslands
(Li et al., 2014). Furthermore, shrublands in TP can have either evergreen leaves or deciduous leaves depending on the local
environments they grow, yet are largely ignored in existing 10 m resolution land cover datasets (Venter et al., 2022). These
unique ecosystems in TP are of high significance to monitor, given that TP has experienced dramatic warming (Fu et al.,
2021), increased humidity (Yang et al., 2014), rapid glacier retreat (Zhao et al., 2022a), permafrost thawing (Gao et al., 2021),
expansion of lakes (Zhang et al., 2020), and vegetation changes (Wang et al., 2020; Duan et al., 2021; Gao et al., 2014) in the
last decades. Thus, a detailed and accurate mapping of the diverse vegetation types in TP is required for understanding climate
change effects on the terrestrial ecosystem, yet is challenging to accomplish given that shrublands are often confused with
forests or alpine meadows and alpine grasslands are commonly misclassified as bare land in most products (Liu et al., 2021;
Cai et al., 2022; Yu et al., 2014). Moreover, the extremely rough terrain in TP results in large mountain shadows and variations

in slope aspects which complicates the accurate detection of vegetation types from satellite imagery (Pizarro et al., 2022; Wang et al., 2023b).

To address the aforementioned challenges, we developed a specific vegetation remote sensing fine classification system tailored for the TP, consisting of 12 vegetation classes and 3 non-vegetation classes. We then created a comprehensive training

and validation dataset consisting of 10,242 samples through manual interpretation and field trips, based on which we performed land cover classification of the TP by integrating multiple data sources on the Google Earth Engine (GEE) platform, including satellite imagery of Sentinel-1 and Sentinel-2, topography, temperature, and precipitation. We investigated the performance of 4 different classification models provided in GEE and selected the highest-accuracy one to generate a 10 m resolution land cover product for the TP in 2022, referred as TP_LC10-2022.

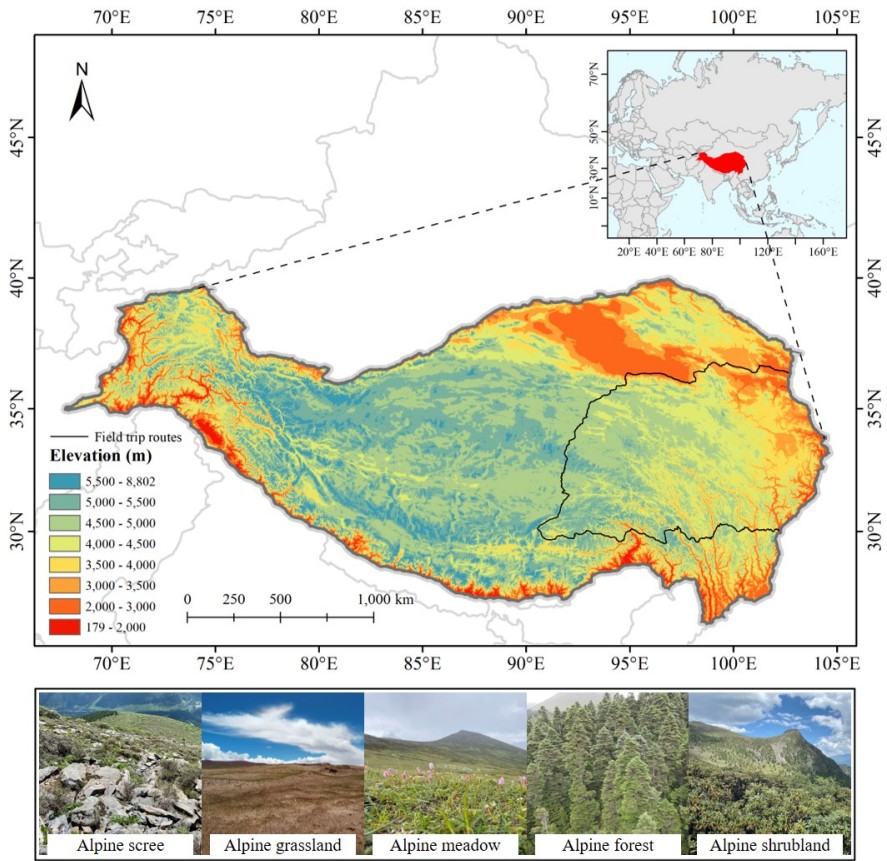

**Figure 1.** Overview of the study area colored by elevation. The black lines are the field trip routes along national roads. The photos show examples of the landscape views of typical vegetation types in the Tibetan Plateau.

## 2 Study Area and Data

### 2.1 Study area

The TP spans from the northern foot of the West Kunlun Mountains and Qilian Mountains to the southern foot of the Himalayas and other mountain ranges, extending from the western edge of the Kunlun Mountains and Pamir Plateau to the eastern edge of the Hengduan Mountains (Fig. 1). It lies between latitudes $25°59'30''$N to $40°1'0''$N and longitudes $67°40'37''$E to $104°40'57''$E, covering a total area of 3.083 million km$^2$. Its average elevation is approximately 4,320 m (Zhang et al., 2022).

Due to the combined influence of climate, topography, and human activities over time, the vegetation cover types vary significantly at different altitudes in the TP. The northwestern and central regions are characterized by extensive bare lands, alpine screes, and persistent snow cover. In the southern and eastern areas, there is a distribution of evergreen forests and mixed forests consisting of needle-leaved and broadleaved trees. The transitional zone between these regions is characterized by shrublands, alpine grasslands, and alpine meadows. We investigated the vegetation cover in a field trip carried out along the national road No. 318 and 109 in July 2023 (Fig. 1), covering all the vegetation types in TP.

### 2.2 Data

#### 2.2.1 Satellite imagery

We used both the optical imagery from Copernicus Sentinel-2 and radar imagery from Copernicus Sentinel-1 for the classification. Sentinel-2 comprises two high-resolution multispectral imaging satellites, each equipped with a multispectral imager. It consists of 13 bands, with spatial resolutions of 10 m for 4 bands, 20 m for 6 bands, and 60 m for 3 bands. The study utilized Level-2A products from the year 2022, which had undergone processing via the Sen2Cor tool at the Copernicus Scientific Data Hub (Doxani et al., 2018). Annual remote sensing images have proven to accurately capture phenological changes in specific vegetation cover and have been successfully utilized in various large-scale land cover classification studies (Verde et al., 2020). Hence, in this study, the Sentinel-2 remote sensing images from the entire year of 2022 were selected for band feature extraction. In this study, the initial step involved retaining the images with a cloud cover of less than 10%. Subsequently, the quality assessment information (QA band) was utilized to exclude pixels with inadequate quality through cloud masking.

Sentinel-1 comprises two polar-orbiting satellites positioned in the same orbital plane. For this research, the Ground Range Detected (GRD) data obtained in wide swath (IW) mode was chosen. The GRD data consists of single polarization (VV) and dual polarization (VV, VH) interferometric wave modes, offering a 10 m resolution (Prats-Iraola et al., 2015). It enables the provision of radar images suitable for land and maritime services, regardless of weather conditions and time of day. The median compositing method in GEE (Souza Jr et al., 2020; Phan et al., 2020) was applied to process all bands of Sentinel-1 and Sentinel-2.

### 2.2.2 Topography data

Shuttle Radar Topography Mission (SRTM) (Farr et al., 2000) was designed to generate high-quality digital elevation models (DEMs) globally using synthetic aperture radar technology. The data collected by SRTM was used to create a global elevation model with a horizontal accuracy of 16 m and vertical accuracy of 6 m, at a spatial resolution of 30 m (Yang et al., 2011).

### 2.2.3 Precipitation data

The Climate Hazards Group InfraRed Precipitation with Station data (CHIRPS) (Funk et al., 2015) is a comprehensive dataset documenting global precipitation from 1981 to the present. CHIRPS integrates satellite imagery with in-situ station data, providing a resolution of $0.05°$ to generate gridded rainfall time-series suitable for trend analysis and seasonal drought monitoring.

### 2.2.4 Temperature data

The ERA5-Land dataset (Muñoz-Sabater et al., 2021) offers a comprehensive reanalysis of land variables, presenting a consistent perspective on their evolution over multiple decades at a higher resolution than ERA5. As the land component of the ECMWF ERA5 climate reanalysis, ERA5-Land combines model data and global observations to create a coherent dataset utilizing the principles of physics. Nineteen extra bands were incorporated by GEE, with each corresponding to an accumulation band, and the hourly values were calculated as the difference between 2 successive forecast steps (Muñoz-Sabater, 2019). For this study, hourly temperature data with a resolution of $0.1°$ from 2022 were used.

## 3 Methodology

### 3.1 Land cover classification

The advancement of cloud computing technology in remote sensing has revolutionized the rapid analysis and application of Earth system science on a large scale, even globally. GEE stands out among these technologies, offering online visualization, computation, and analysis capabilities for extensive Earth science data (Gorelick et al., 2017; Kumar and Mutanga, 2018). Consequently, we opted to utilize GEE for data processing, and analysis. Importantly, the satellite data and auxiliary data relevant to this study can be readily accessed through GEE. Fig. 2 presents our comprehensive classification system, which comprises 4 main steps: 1) sampling strategy, 2) data preprocessing and feature construction, 3) classification model comparison, and 4) accuracy assessment and inter-comparison.

### 3.1.1 Classification system

The TP harbors the world's highest and one of the most distinctive alpine vegetation communities, which pose challenges to their inclusion in both global and Chinese land cover classification systems. To address this issue, we have developed an adapted classification system specifically tailored to the alpine vegetation types found in the TP. The basis for constructing this classification system are as follows:

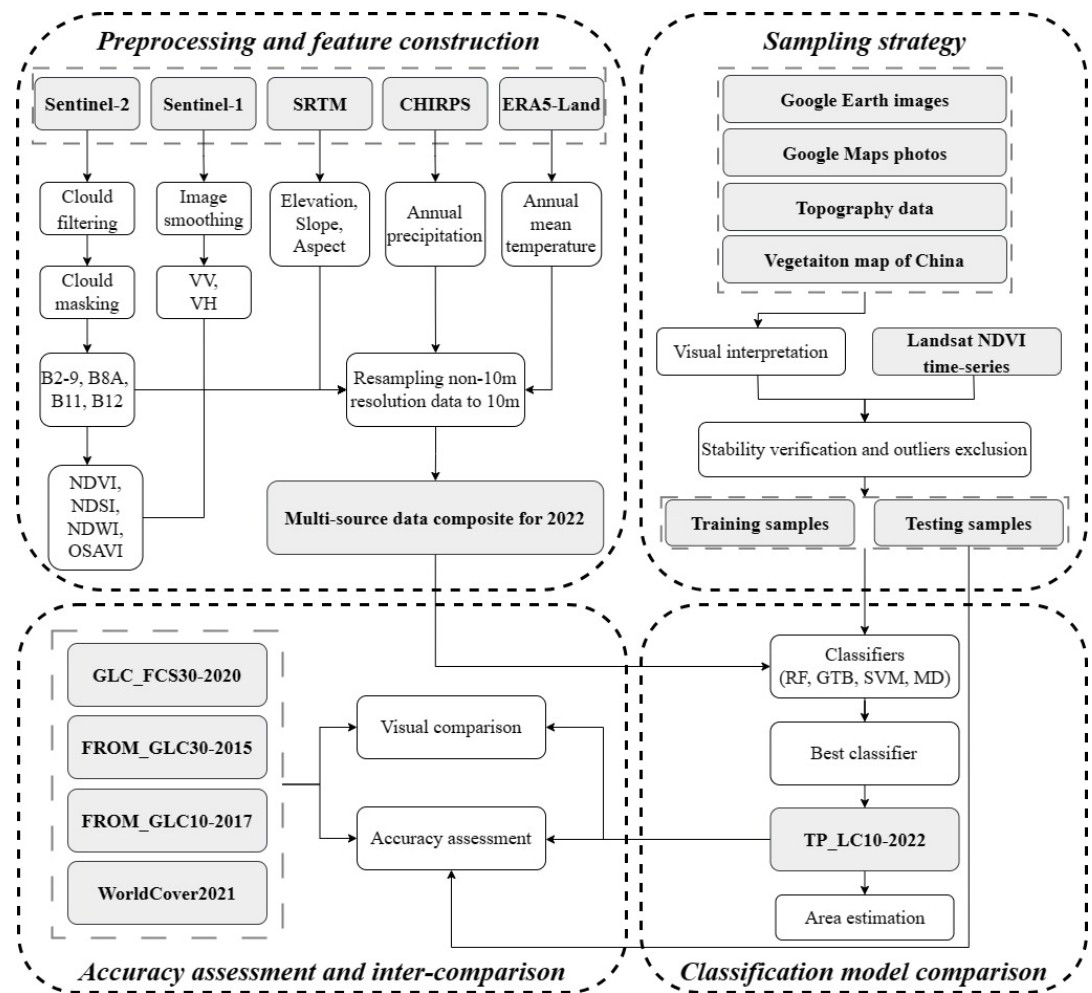

**Figure 2.** Flowchart of the land cover classification carried out in this study.

(1) Comprehensive vegetation functional types: We have categorized the vegetation in the TP based on plant growth form (trees, shrubs, and herbs), leaf phenology (evergreen and deciduous), leaf type (broadleaved and needle-leaved), and ecosystem type. This classification system results in 12 vegetation types, including 5 types of tree cover including evergreen needle-leaved forest (ENF), deciduous needle-leaved forest (DNF), evergreen broadleaved forest (EBF), deciduous broadleaved forest (DBF), and mixed forest (MF); 2 types of shrub cover including evergreen shrubland (ES) and deciduous shrubland (DS); 2 types of herb cover including alpine grassland (AG) and alpine meadow (AM); 3 special vegetation cover types including alpine scree (AS), wetland (WL), and cultivated vegetation (CV); and 3 non-vegetation land cover types, including bare land (BL), water body (WB), and permanent ice and snow (PIS).

(2) Discriminability of different vegetation functional types in remote sensing imagery: During the classification stage, we can effectively differentiate various land cover types, including diverse vegetation, utilizing the discriminative capabilities of

the multispectral bands of Sentinel-2 (Liu et al., 2023b). Moreover, the incorporation of high-resolution Google Earth imagery, with a spatial resolution of up to 0.3 meters, enhances the distinguishability of land cover types during the sample selection phase. This ensures the feasibility of visually interpreting large-scale samples from remote sensing imagery and obtaining reliable and up-to-date information (Gong et al., 2013).

In this study, we did not specifically select samples of built-up areas and instead categorized bare land together with built-up areas for two primary reasons. Firstly, built-up areas account for only 0.092% of the total area in ESA WorldCover2021, highlighting their relatively small extent compared to other land cover types (Zanaga et al., 2022). Secondly, bare land in our product exhibits spectral characteristics similar to those of built-up areas, resulting in the classification of most built-up areas as bare land (Li et al., 2017).

### 3.1.2 Sampling strategy

Supervised classification models heavily depend on a substantial number of labeled samples for effective training and validation (Foody and Mathur, 2004). While extracting samples directly from existing land cover products can save manpower, it introduces several issues: (1) Extracted training samples may inherit errors from previous land cover products (Xi et al., 2022); (2) Utilizing low-resolution products to extract training samples for high-resolution land cover mapping can lead to information loss and boundary effects between adjacent land parcels (Zhang et al., 2021b; Zhang and Roy, 2017); (3) Reconciling classification systems of different products is difficult, and global land cover products may not include specific land cover types for certain regions. Therefore, collecting samples through visual interpretation emerges as a more feasible approach (Schepaschenko et al., 2019).

Google Earth integrates high-resolution imagery from sources like QuickBird and GeoEye, providing reliable remote sensing data sources for visual interpretation. Selecting samples in areas without Google Earth image coverage in 2022 poses a challenge. Normalized Difference Vegetation Index (NDVI) time-series have thus been used as auxiliary data for land cover sample selection (Yang and Huang, 2021; Feng et al., 2016). To ensure the selection of stable samples, this study examines the stability of land features by reviewing the Landsat NDVI time-series from 2013 to 2022. To eliminate the interference of clouds and snow in the NDVI time-series, the following operations were performed on Landsat images: 1) Filtering out pixels with cloud coverage greater than 50%; 2) When selecting forest and shrub samples, applying a Normalized Difference Snow Index (NDSI) mask to filter out pixels with NDSI greater than -0.4. To obtain a more continuous NDVI time-series, the harmonic analysis of time series (HANTS) model was used for data interpolation and smoothing to remove noise and reconstruct missing data (Zhou et al., 2015). By following the steps outlined above, we detected land cover changes during 2013-2022 using Landsat NDVI time series (Fig. A2). This approach helps to avoid selecting sites where land cover change has occurred. Additionally, the monthly mean value of the NDVI time-series for 2013-2022 was calculated to determine the phenological characteristics of each sample point (Chu et al., 2021). All samples were interpreted based on Google Earth images, with subsequent verification using NDVI time series as a supplementary measure to ensure stability and detect phenology.

For instance, in Fig. 3, different color characteristics are observed for evergreen shrubs and deciduous shrubs in Google Earth imagery. Evergreen shrubs maintain their green color even during winter, while deciduous shrubs appear yellow-brown.

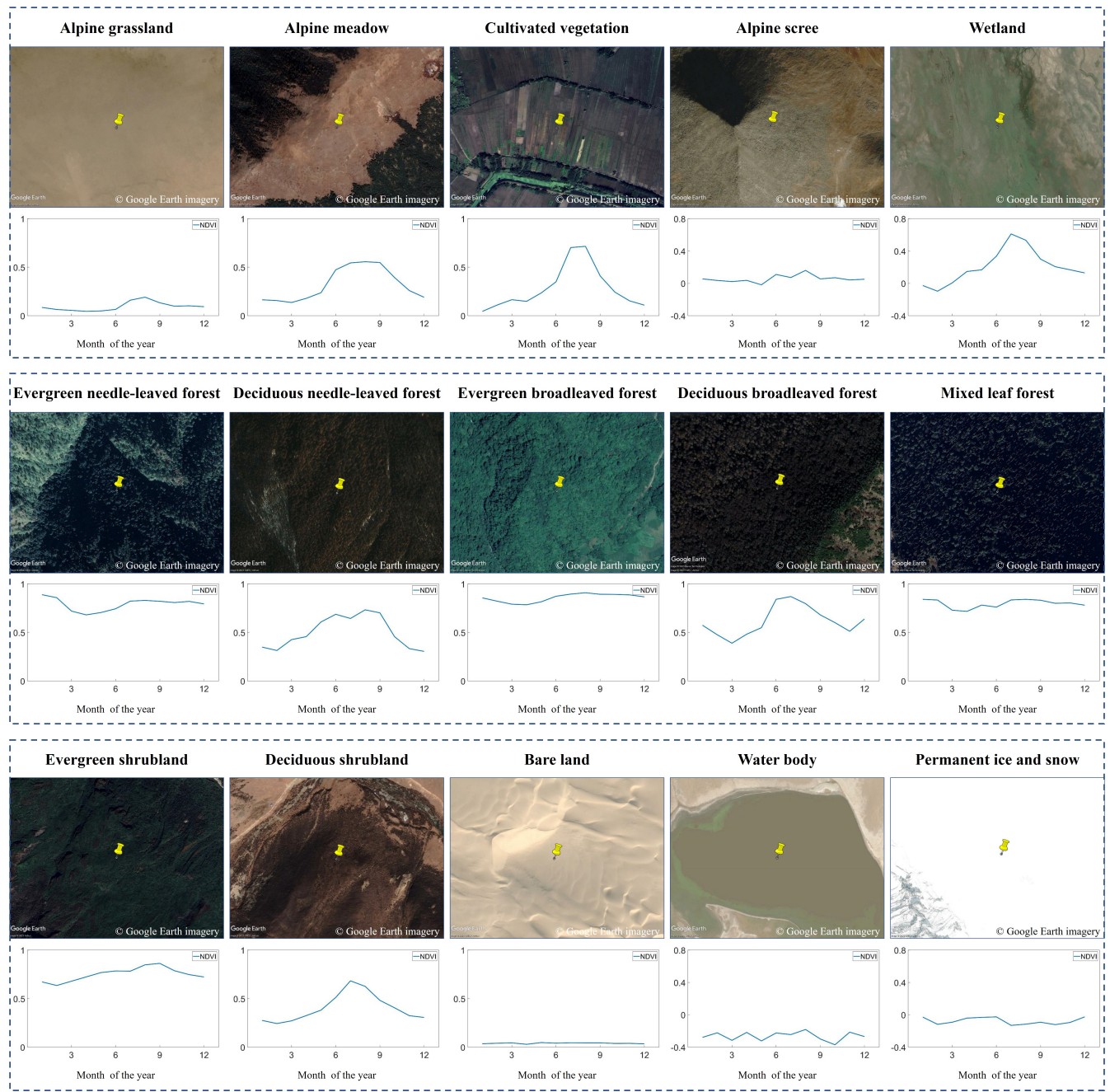

**Figure 3.** Examples of auxiliary data used for explaining visual interpretation, including Google Earth imagery and the Landsat monthly mean Normalized Difference Vegetation Index (NDVI) time-series for 2013-2022. The x-axis represents month of the year and the y-axis represents NDVI value.

**Table 1.** Number of training and validation samples for the 15 land cover types.

| Land cover type | Number of training samples | Number of validation samples | Total |
|---|---|---|---|
| Bare land | 883 | 234 | 1117 |
| Alpine scree | 583 | 129 | 712 |
| Alpine grassland | 607 | 149 | 756 |
| Alpine meadow | 949 | 226 | 1175 |
| Evergreen needle-leaved forest | 654 | 169 | 823 |
| Deciduous needle-leaved forest | 432 | 106 | 538 |
| Evergreen broadleaved forest | 550 | 130 | 680 |
| Deciduous broadleaved forest | 459 | 132 | 591 |
| Mixed forest | 215 | 78 | 293 |
| Evergreen shrubland | 566 | 153 | 719 |
| Deciduous shrubland | 663 | 157 | 820 |
| Water body | 504 | 114 | 618 |
| Wetland | 280 | 72 | 352 |
| Cultivated vegetation | 441 | 91 | 532 |
| Permanent ice and snow | 414 | 102 | 516 |
| Total | 8200 | 2042 | 10242 |

However, during spring or summer, direct differentiation between the two from imagery is not possible. Therefore, phenological characteristics are extracted from their mean NDVI time-series. Evergreen shrubs exhibit relatively stable NDVI values, whereas deciduous shrubs show a decrease in NDVI due to seasonal leaf shedding. Evergreen needle-leaved forests, evergreen broadleaved forests, and evergreen shrublands exhibit similar trends and values in NDVI time series. However, they can be discerned in Google Earth images based on their distinctive crown shapes and textures (Fig. 3).

Google Earth imagery does not accurately determine the presence of herbaceous plant growth. Nevertheless, grasslands display a significant periodic increase in NDVI during the growing season, while bare land exhibit a relatively flat NDVI time-series. This characteristic is utilized for identifying bare land. Regarding alpine grasslands and alpine meadows, judgments are based on area size, vegetation composition, moisture condition, and terrain. Meadows typically have a smaller area compared to grasslands, better moisture conditions and are often accompanied by trees or shrubs in the vicinity. Grasslands have a flatter distribution area compared to meadows, as depicted in Fig. 3. Consequently, effective differentiation between alpine grasslands and alpine meadows is achieved.

Topography data (elevation, slope, aspect) (Farr et al., 2000), the 1:1 million Chinese vegetation map (Su et al., 2020), and high-quality Google Maps photos were selected for auxiliary judgment. Ultimately, a total of 10,242 samples were collected, as illustrated in Fig. 4. Subsequently, the 10,242 samples were mixed, and the samples for each category were randomly divided

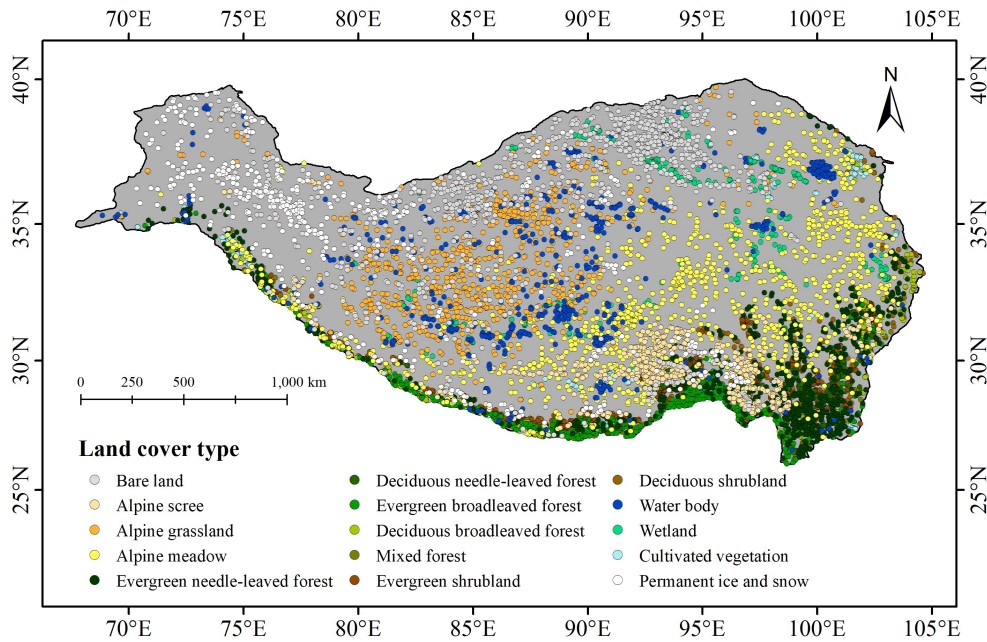

**Figure 4.** Spatial distribution of the 10, 242 samples for land cover classification in the Tibetan Plateau.

into training and validation sets in an approximate 4:1 ratio, as presented in Table 1. We adjusted the ratio of training to validation samples to 4:1 instead of the commonly used 7:3 to enhance the model's fitting capability to handle the complex distribution of features (Ramezan et al., 2021).

### 3.1.3 Feature construction for classification

The selected input bands for Sentinel-2 included B2-B8, B8A, B9, B11, and B12. Among these bands, B2-B8, B11, and B12 have been demonstrated to be effective in classifying deciduous and coniferous tree species (Immitzer et al., 2016; Li et al., 2021a). Additionally, B8A is suitable for boreal landscape classification (Abdi, 2020), while B9 values demonstrate differences between bare soil and vegetation-covered areas (Zhao et al., 2023b), making them useful for classification purposes. For Sentinel-1 images, utilizing both VV and VH can enhance classification accuracy, leading to their selection as input features (Jacob et al., 2020; Steinhausen et al., 2018).

To better discern the characteristics of various land features, we calculated several indices using Sentinel-2 imagery. These included the NDVI, NDSI, Normalized Difference Water Index (NDWI), and Optimized Soil-Adjusted Vegetation Index (OS-AVI). NDVI is highly sensitive to vegetation growth and is commonly used to distinguish between vegetated and non-vegetated areas (Rouse et al., 1974). NDSI effectively detects snow by utilizing the reflective properties of snow in the short infrared band, making it advantageous for studying ice and snow coverage in high mountain regions (Dozier, 1989). NDWI effectively

distinguishes between water and non-water features (Xu, 2006). OSAVI improves the sensitivity and stability of vegetation indices by considering the influence of soil reflectance, providing a more accurate reflection of vegetation coverage and growth conditions, particularly in cases of bare soil or sparse vegetation (Rondeaux et al., 1996).

Topography significantly influence the vertical distribution of vegetation in high mountain areas (Zou et al., 2023). Therefore, in this study, we included elevation, slope, and aspect as input features for classification. Additionally, we incorporated annual precipitation and mean annual temperature as classification feature indicators (Wang et al., 2023a; Shen et al., 2015). For bands with a spatial resolution different from 10 m, we employed bicubic interpolation to resample them to 10 m resolution for mapping (Liu et al., 2020). All the features and their detailed descriptions are presented in Table 2.

**Table 2.** Features used for land cover classification.

| Data source | Feature | Description |
| --- | --- | --- |
| Sentinel-1 | VV | Single co-polarization, vertical transmit/vertical receive, descending orbit |
| | VH | Dual-band cross-polarization, vertical transmit/horizontal receive, descending orbit |
| Sentinel-2 | B2 | Blue Band Reflectance ($Blue$) |
| | B3 | Green Band Reflectance ($Green$) |
| | B4 | Red Band Reflectance ($Red$) |
| | B5 | Vegetation Red-Edge 1 Band Reflectance |
| | B6 | Vegetation Red-Edge 2 Band Reflectance |
| | B7 | Vegetation Red-Edge 3 Band Reflectance |
| | B8 | Near-Infrared Band Reflectance ($NIR$) |
| | B8A | Narrow Near-Infrared Band Reflectance |
| | B9 | Water Vapor Band Reflectance |
| | B11 | Shortwave Infrared 1 Band Reflectance ($SWIR1$) |
| | B12 | Shortwave Infrared 2 Band Reflectance ($SWIR2$) |
| | NDVI | $NDVI = (NIR - Red)/(NIR + Red)$ |
| | DNSI | $NDSI = (Green - SWIR1)/(Green + SWIR1)$ |
| | NDWI | $NDWI = (Green - NIR)/(Green + NIR)$ |
| | OSAVI | $OSAVI = (NIR - Red)/(NIR + Red + 0.16)$ |
| SRTM | Elevation | |
| | Slope | |
| | Aspect | |
| CHIRPS | Annual precipitation | |
| ERA5-Land | Annual mean temperature | Temperature of air at 2m above the surface of land or in-land waters. |

### 3.1.4 Classification models comparison

Machine learning is a typically employed technique in remote sensing image classification. To identify the most appropriate classification model, we compared 4 widely-used machine learning models in GEE, including Random Forest (RF) (Breiman, 2001), Gradient Tree Boosting (GTB) (Friedman, 2001), Support Vector Machine (SVM) (Hearst et al., 1998), and Minimum Distance (MD) (Wacker and Landgrebe, 1972). We fine-tuned the parameters of all the classification models to achieve optimal results (Table A1). The classification model with the highest overall performance was chosen to generate the land cover map and calculate the area proportion of each land cover type.

### 3.1.5 Accuracy assessment and inter-comparison

The accuracy of remote sensing image classification is commonly assessed using a confusion matrix, which provides 4 quantitative indicators: Producer's Accuracy (P.A.) for measuring omission errors, User's Accuracy (U.A.) for measuring commission errors, Overall Accuracy (O.A.), and Kappa coefficient.

To compare with existing 4 global land cover datasets, namely ESA WorldCover2021, FROM_GLC10-2017, FROM_GLC30-2015, and GLC_FCS30-2020, we merged pixels belonging to the same class (Table A2) and employed randomly sampled validation samples. Additionally, we selected three $0.1° \times 0.1°$ grids within the TP to compare the visual classification results of TP_LC10-2022 with the existing 4 land cover products.

## 4 Results and discussion

### 4.1 Comparison of classification models

Table 3 presents the evaluation results of different classification models applied in the study area using GEE. The results demonstrate that the RF model achieved the highest accuracy, with an Overall Accuracy (O.A.) of 86.5% and a Kappa coefficient of 0.854. The Gradient Tree Boosting (GTB) model closely followed with an O.A. of 85.6% and a Kappa coefficient of 0.844. The Minimum Distance (MD) model yielded an accuracy of O.A. 79.7% and Kappa 0.781, while the Support Vector Machine (SVM) exhibited significantly lower classification results, with an O.A. of 64.7% and Kappa of 0.618.

The high accuracy achieved by RF and GTB models can be attributed to their ensemble learning algorithms based on decision trees. These algorithms combine multiple decision trees to enhance model performance and generalization capabilities (Salditt et al., 2022). In contrast to the findings of Abdi (2020), where RF and SVM exhibited similar O.A., our SVM showed a decline of 21.8% compared to RF (Tu et al., 2020). This discrepancy may be attributed to RF's ability to mitigate the correlation between samples and features through random sampling and feature selection, resulting in improved classification performance and robustness. Moreover, RF can effectively handle high-dimensional data and capture nonlinear relationships by integrating multiple decision trees (Tu et al., 2020; Gislason et al., 2006).

The three classification models, excluding SVM, effectively distinguished water bodies from other land cover types, achieving a P.A. exceeding 0.99. However, all classification models performed fairly in differentiating mixed forests. For instance,

**Table 3.** Comparison of classification results for Random Forest (RF), Gradient Tree Boosting (GTB), Minimum Distance (MD), and Support Vector Machine (SVM) at their best performance. Bold font denotes highest U.A. and P.A. within each land cover type, as well as the highest O.A. and Kappa among the 4 models.

| | | BL | AS | AG | AM | ENF | DNF | EBF | DBF | MF | ES | DS | WB | WL | CV | PIS | O.A. | Kappa |
|---|---|---|---|---|---|---|---|---|---|---|---|---|---|---|---|---|---|---|
| RF | P.A. | 0.974 | **0.884** | **0.953** | **0.894** | 0.805 | **0.830** | **0.915** | 0.750 | 0.462 | **0.856** | **0.847** | **1.000** | 0.611 | **0.923** | 0.941 | **0.865** | **0.854** |
| | U.A. | **0.942** | 0.851 | **0.953** | 0.831 | **0.764** | 0.871 | 0.815 | **0.786** | **0.800** | 0.851 | **0.821** | **0.983** | **0.898** | 0.884 | **0.941** | | |
| GTB | P.A. | **0.979** | 0.868 | 0.919 | **0.894** | 0.769 | 0.774 | 0.892 | **0.788** | **0.513** | 0.843 | 0.834 | 0.991 | **0.639** | **0.923** | 0.902 | 0.856 | 0.844 |
| | U.A. | **0.942** | **0.855** | 0.951 | **0.835** | 0.756 | 0.872 | 0.835 | 0.759 | 0.741 | 0.838 | 0.775 | **0.983** | 0.885 | **0.894** | 0.902 | | |
| SVM | P.A. | 0.688 | 0.791 | 0.584 | 0.681 | 0.396 | 0.717 | 0.646 | 0.636 | 0.269 | 0.621 | 0.643 | 0.833 | 0.597 | 0.780 | 0.784 | 0.647 | 0.618 |
| | U.A. | 0.703 | 0.729 | 0.561 | 0.661 | 0.429 | 0.623 | 0.587 | 0.592 | 0.447 | 0.674 | 0.574 | 0.969 | 0.642 | 0.664 | 0.930 | | |
| MD | P.A. | 0.885 | 0.814 | 0.711 | 0.863 | **0.876** | 0.660 | 0.777 | 0.780 | 0.397 | 0.745 | 0.771 | **1.000** | 0.625 | 0.758 | **0.971** | 0.797 | 0.781 |
| | U.A. | 0.885 | 0.766 | 0.914 | 0.739 | 0.643 | **0.946** | **0.863** | 0.665 | 0.674 | **0.891** | 0.742 | 0.950 | 0.750 | 0.852 | 0.846 | | |

BL: bare land; AS: alpine scree; AG: alpine grassland; AM: alpine meadow; ENF: evergreen needle-leaved forest; DNF: deciduous needle-leaved forest;

EBF: evergreen broadleaved forest; DBF: deciduous broadleaved forest; MF: mixed forest; ES: evergreen shrubland; DS: deciduous shrubland;

WB: water body; WL: wetland; CV: cultivated vegetation; PIS: permanent ice and snow

SVM achieved a low P.A. of only 0.269 for mixed forest classification. Despite the integration of various machine learning models within GEE, including algorithms like RF, a distinct absence of direct support for deep learning persists. This is notable even in light of the well-established and showcased capabilities of deep learning in the fine-grained classification of land cover (Wang et al., 2023c). This limitation, to a certain extent, poses a hindrance to the extensive application of large-scale land cover mapping.

The utilization of multi-source remote sensing data can offer a more comprehensive understanding of land cover (Xu et al., 2022; Chen et al., 2017). Given to the importance of features, all features contributed to the mapping and elevation contributed slightly more to the accuracy of the classification (Fig. A1). This is attributed to the impact of the TP's rugged terrain on the hydrothermal conditions in distinct regions, leading to notable variations in vegetation phenology (Hwang et al., 2011; Sang et al., 2024).

## 4.2 Land cover classification map

Fig. 5a provides an overview of the TP_LC10-2022 product and 4 global land cover products, along with the proportion of each land cover type in TP_LC10-2022. Alpine meadow and alpine grassland account for the proportions at 23.76% and 16.48%, respectively. Alpine scree surprisingly ranks fourth, with a proportion of 13.99%, after alpine meadow, bare land, and alpine grassland. Evergreen needle-leaved forest has the largest area among the forest types, and deciduous shrubland has a larger area than evergreen shrubland, reaching 3.57%, surpassing other forest types except for evergreen needle-leaved forest. Table A4 presents the statistical area results of 5 land cover products in the TP, highlighting significant discrepancies among them.

According to Fig. 5b, ESA WorldCover2021, FROM_GLC10-2017, and FROM_GLC30-2015 products overestimate the area of bare land in the TP, similar to the issues observed in FROM_GLC-agg and ESA CCI land cover products (Liu et al., 2021; Yu et al., 2014). This may be due to the misclassification of alpine grassland as bare land because these products captured

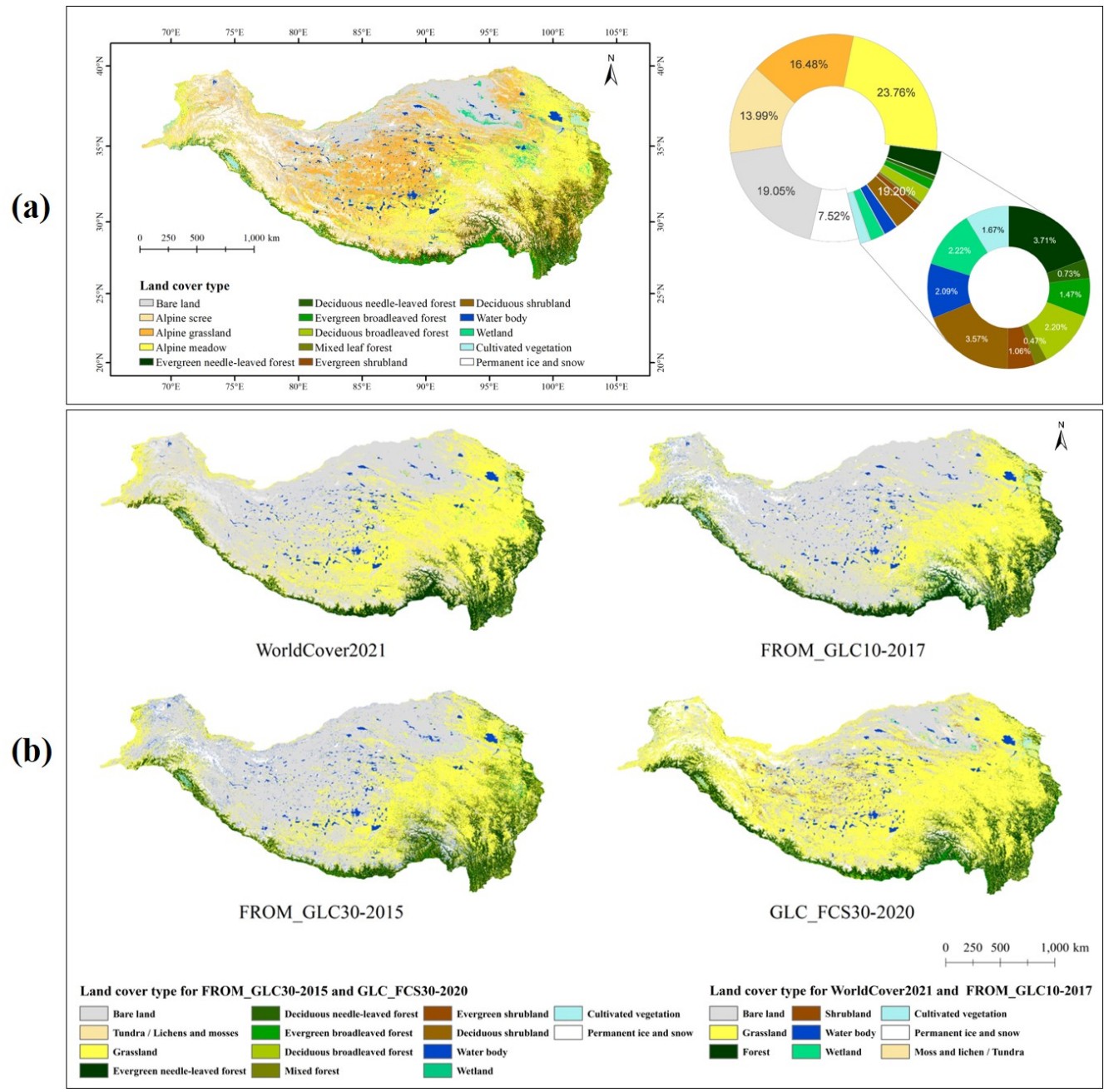

**Figure 5.** The overall Comparison for TP_LC10-2022 and other land cover maps. (a) TP_LC10-2022 and the proportion of 15 land cover types. (b) An overview of 4 land cover products in the Tibetan Plateau, including ESA WorldCover2021, FROM_GLC10-2017, FROM_GLC30-2015 and GLC_FCS30-2020. Legend fusion rules for WorldCover2021 and FROM_GLC10-2017 are provided in Table A2, and for FROM_GLC30-2015 and GLC_FCS30-2020, refer to Table A3.

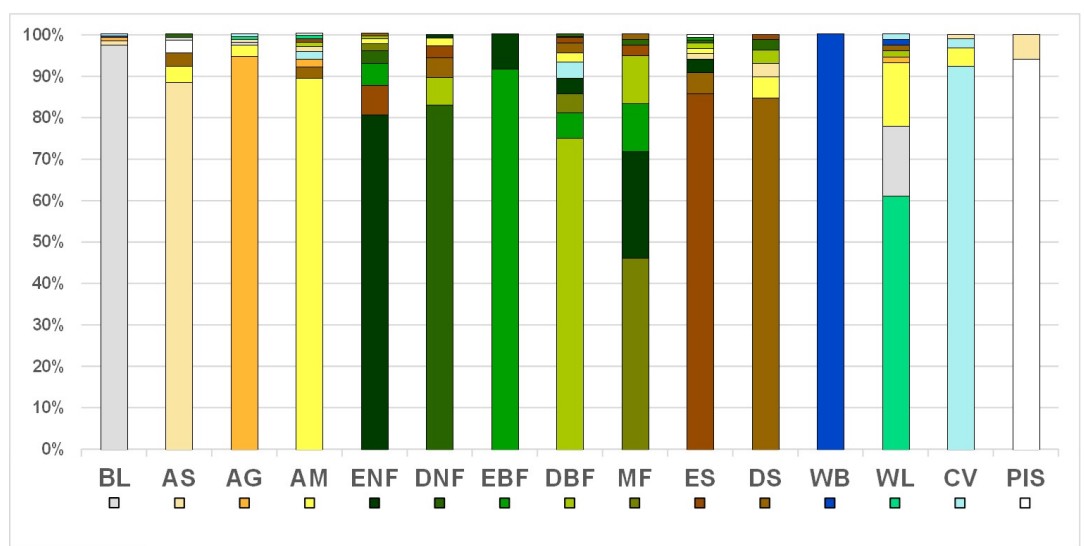

**Figure 6.** The confusion proportions for each of the land cover types in TP_LC10-2022. BL: bare land; AS: alpine scree; AG: alpine grassland; AM: alpine meadow; ENF: evergreen needle-leaved forest; DNF: deciduous needle-leaved forest; EBF: evergreen broadleaved forest; DBF: deciduous broadleaved forest; MF: mixed forest; ES: evergreen shrubland; DS: deciduous shrubland; WB: water body; WL: wetland; CV: cultivated vegetation; PIS: permanent ice and snow.

less spectral information during the growing season of alpine grasslands. GLC_FCS30-2020 exhibits the highest consistency with TP_LC10-2022 regarding bare land (Table A4 and Fig. 5) and it classified more grasslands while failed to differentiate between grasslands and meadows. Additionally, GLC_FCS30-2020 assigns 61.44% of the total TP area as grassland, indicating an overestimation of grassland extent (Table A4).

The TP exhibits significant variations in annual rainfall and land surface temperature across its diverse regions, resulting in distinct hot and cold spots (Rao et al., 2019; Wu et al., 2019). Consequently, leveraging climate data can prove beneficial in categorizing alpine meadows in the southeastern TP and alpine grasslands in the northwestern TP at regional climatic scales, given their high sensitivity to changes in annual precipitation and land surface temperature (Su et al., 2020; Wang et al., 2021). Our study also found incorporating resampled coarse-resolution climate data can help improve the classification accuracy of

finer resolution data (Jia et al., 2014). However, it may cause potential loss of spatial information (Xu et al., 2020), which has not been observed in the TP_LC10-2022 dataset.

Table 4 illustrates the confusion matrix of TP_LC10-2022, with an overall accuracy of 86.5% and a Kappa coefficient of 0.854. Water body achieved a P.A. of 100%, while mixed forest only reached 46.2%. Fig. 6 shows that most mixed forests are challenging to differentiate from other forest types, with over 25% of mixed forests misclassified as evergreen needle-leaved

forests and 11.5% misclassified as evergreen broadleaved forests or deciduous broadleaved forests. The classification accuracy of wetlands is also unsatisfactory, with a P.A. of only 61.1%. Over 16% of wetlands were classified as bare land, and over 15%

**Table 4.** Confusion matrix of TP_LC10-2022 product extracted using Random Forest (RF) classification model. Bold font denotes correctly classified sample points.

| | BL | AS | AG | AM | ENF | DNF | EBF | DBF | MF | ES | DS | WB | WL | CV | PIS | Total | P.A. |
|---|---|---|---|---|---|---|---|---|---|---|---|---|---|---|---|---|---|
| BL | **228** | 2 | 2 | 0 | 0 | 0 | 0 | 0 | 0 | 0 | 0 | 1 | 0 | 1 | 0 | 234 | 0.974 |
| AS | 1 | **114** | 0 | 5 | 0 | 1 | 0 | 0 | 0 | 0 | 4 | 0 | 0 | 0 | 4 | 129 | 0.884 |
| AG | 1 | 1 | **141** | 4 | 0 | 0 | 0 | 0 | 0 | 0 | 0 | 0 | 1 | 1 | 0 | 149 | 0.953 |
| AM | 0 | 3 | 4 | **202** | 0 | 0 | 0 | 2 | 0 | 2 | 6 | 0 | 2 | 4 | 1 | 226 | 0.894 |
| ENF | 0 | 0 | 0 | 2 | **136** | 5 | 9 | 1 | 3 | 12 | 1 | 0 | 0 | 0 | 0 | 169 | 0.805 |
| DNF | 0 | 0 | 0 | 2 | 1 | **88** | 0 | 7 | 0 | 3 | 5 | 0 | 0 | 0 | 0 | 106 | 0.830 |
| EBF | 0 | 0 | 0 | 0 | 11 | 0 | **119** | 0 | 0 | 0 | 0 | 0 | 0 | 0 | 0 | 130 | 0.915 |
| DBF | 0 | 0 | 0 | 3 | 5 | 1 | 8 | **99** | 6 | 2 | 3 | 0 | 0 | 5 | 0 | 132 | 0.750 |
| MF | 0 | 0 | 0 | 0 | 20 | 1 | 9 | 9 | **36** | 2 | 1 | 0 | 0 | 0 | 0 | 78 | 0.462 |
| ES | 0 | 2 | 0 | 2 | 5 | 1 | 1 | 2 | 0 | **131** | 8 | 0 | 0 | 0 | 1 | 153 | 0.856 |
| DS | 0 | 5 | 0 | 8 | 0 | 4 | 0 | 5 | 0 | 2 | **133** | 0 | 0 | 0 | 0 | 157 | 0.847 |
| WB | 0 | 0 | 0 | 0 | 0 | 0 | 0 | 0 | 0 | 0 | 0 | **114** | 0 | 0 | 0 | 114 | 1.000 |
| WL | 12 | 0 | 1 | 11 | 0 | 0 | 0 | 1 | 0 | 0 | 1 | 1 | **44** | 1 | 0 | 72 | 0.611 |
| CV | 0 | 1 | 0 | 4 | 0 | 0 | 0 | 0 | 0 | 0 | 0 | 0 | 2 | **84** | 0 | 91 | 0.923 |
| PIS | 0 | 6 | 0 | 0 | 0 | 0 | 0 | 0 | 0 | 0 | 0 | 0 | 0 | 0 | **96** | 102 | 0.941 |
| Total | 242 | 134 | 148 | 243 | 178 | 101 | 146 | 126 | 45 | 154 | 162 | 116 | 49 | 96 | 102 | 2042 | |
| U.A. | 0.942 | 0.851 | 0.953 | 0.831 | 0.764 | 0.871 | 0.815 | 0.786 | 0.800 | 0.851 | 0.821 | 0.983 | 0.898 | 0.884 | 0.941 | | |
| O.A. | | | | | | | | 0.865 | | | | | | | | | |
| Kappa | | | | | | | | 0.854 | | | | | | | | | |

BL: bare land; AS: alpine scree; AG: alpine grassland; AM: alpine meadow; ENF: evergreen needle-leaved forest; DNF: deciduous needle-leaved forest;

EBF: evergreen broadleaved forest; DBF: deciduous broadleaved forest; MF: mixed forest; ES: evergreen shrubland; DS: deciduous shrubland;

WB: water body; WL: wetland; CV: cultivated vegetation; PIS: permanent ice and snow

were incorrectly classified as alpine meadows. The U.A. for the water body reached 98.3%, while evergreen needle-leaved forests had the lowest U.A. at 76.4%.

In addition, the spectral variations within urban areas have also resulted in substantial uncertainties. Our approach of categorizing built-up areas and bare land may lead to misclassification of urban pixels. To minimize the uncertainties in urban areas on our final map, we applied the ESRI land cover map in 2022 to mask off urban pixels (Karra et al., 2021).

Although we employed the Sentinel-2 median composition method in this study, we acknowledge the potential enhancement that time-series analysis could bring to our research. In comparison to median composition, time-series analysis has the potential to more comprehensively capture phenological information of vegetation, thereby yielding more accurate land cover classification results (Xie et al., 2019; Nguyen et al., 2020). However, time-series methods also have their limitations, such as the requirement for a greater number of valid observations (Hemmerling et al., 2021). For example, during the summer of 2022 (June-August), when setting the "CLOUDY_PIXEL_PERCENTAGE" parameter to 10%, 20%, 30%, and 40%, and applying QA band masking, we lost 13.59%, 5.81%, 2.44%, and 1.32% of the Sentinel-2 image area in the TP. The removed pixels are concentrated mainly in the cloudy southeastern TP (only shown for 10% threshold in Fig. A3) (Tang et al., 2022). This

constraint can preclude the attainment of desired outcomes in regions where cloud-free image availability is low (Chu et al., 2021; Coluzzi et al., 2018).

The blue, red-edge, and shortwave infrared (SWIR) bands of mono-temporal median Sentinel-2 imagery have proven effective for vegetation classification, distinguishing between crop types and tree species (Immitzer et al., 2016). As shown in Fig. A4, both evergreen and deciduous vegetation exhibit similar trends in Sentinel-2 multispectral bands, yet they display

significant differences in spectral reflectance values. This indicates that median composited bands of Sentinel-2, along with constructed spectral indices, can be used to distinguish between evergreen and deciduous vegetation. Median composites are affected by the number of available images, we thus ensured a minimum of three high-quality observations across the entire TP while preprocessing the annual Sentinel-2 images. The composites from $\geq$ three Sentinel images make it possible to achieve the seamless effect shown in Figure A4 in various locations over large areas of the TP. The integration of multiple satellite

images over time helps capture the phenology of different vegetation types while mitigating the influence of outliers (Carrasco et al., 2019; Pizarro et al., 2022; Tu et al., 2020; Verde et al., 2020; Xie et al., 2019).

However, relying solely on median composited bands of Sentinel-2 and constructed spectral indices may not suffice to achieve high classification accuracy, emphasizing the importance of multisource data. Notably, elevation emerges as the most important feature among all ancillary ones (Fig. A1), reflecting the natural distribution of vegetation types, which are predom-

295 inantly shaped by latitudinal zonation in the mountainous TP (Sherman et al., 2008) (Fig. A5 and Fig. 7). Conversely, in flat areas where vegetation distribution is minimally influenced by topography, or in urban areas where vegetation distribution is affected by anthropogenic activity, topographic information may exhibit limitations in land cover classification (Zeng et al., 2019). Thus, leveraging features derived from multisource data allows us to amplify and capture differences between evergreen and deciduous vegetation, as well as between shrubs and woodlands, ultimately leading to a high classification accuracy (Xu

et al., 2018; Yan et al., 2023).

## 4.3 Inter-comparison with other products

The land cover samples selected remained stable encompassing the years from 2013 to 2022 for all the other 4 land cover products, thus making them comparable to our TP_LC10-2022 map. Therefore, we validated the aggregation of samples into 8 categories and assessed the performance of TP_LC10-2022 and the 4 other land cover products in the TP region, as depicted

in Table 5.

For shrubland, the classification performance of the 4 global land cover products is remarkably low. Notably, ESA World-Cover2021 achieves a P.A. and U.A. of 0 for shrubland classification. Among these land cover products, FROM_GLC30-2015 exhibits the highest U.A. for shrubland classification, albeit at a mere 59.3%. This suggests substantial shortcomings in the precise classification of shrubland in the TP region by the current land cover products.

A simultaneous visual comparison was conducted among the 5 products. In Fig. 7a, TP_LC10-2022 and FROM_GLC30-2015 exhibited superior performance, revealing more intricate forest details compared to other products. Notably, other products largely disregarded vast areas of high-elevation alpine shrublands above the timberline, while TP_LC10-2022 delineated them (shown in brown) and exhibited distinct vertical zonation. In Fig. 7b, the other 4 products tended to misclassify shrub-

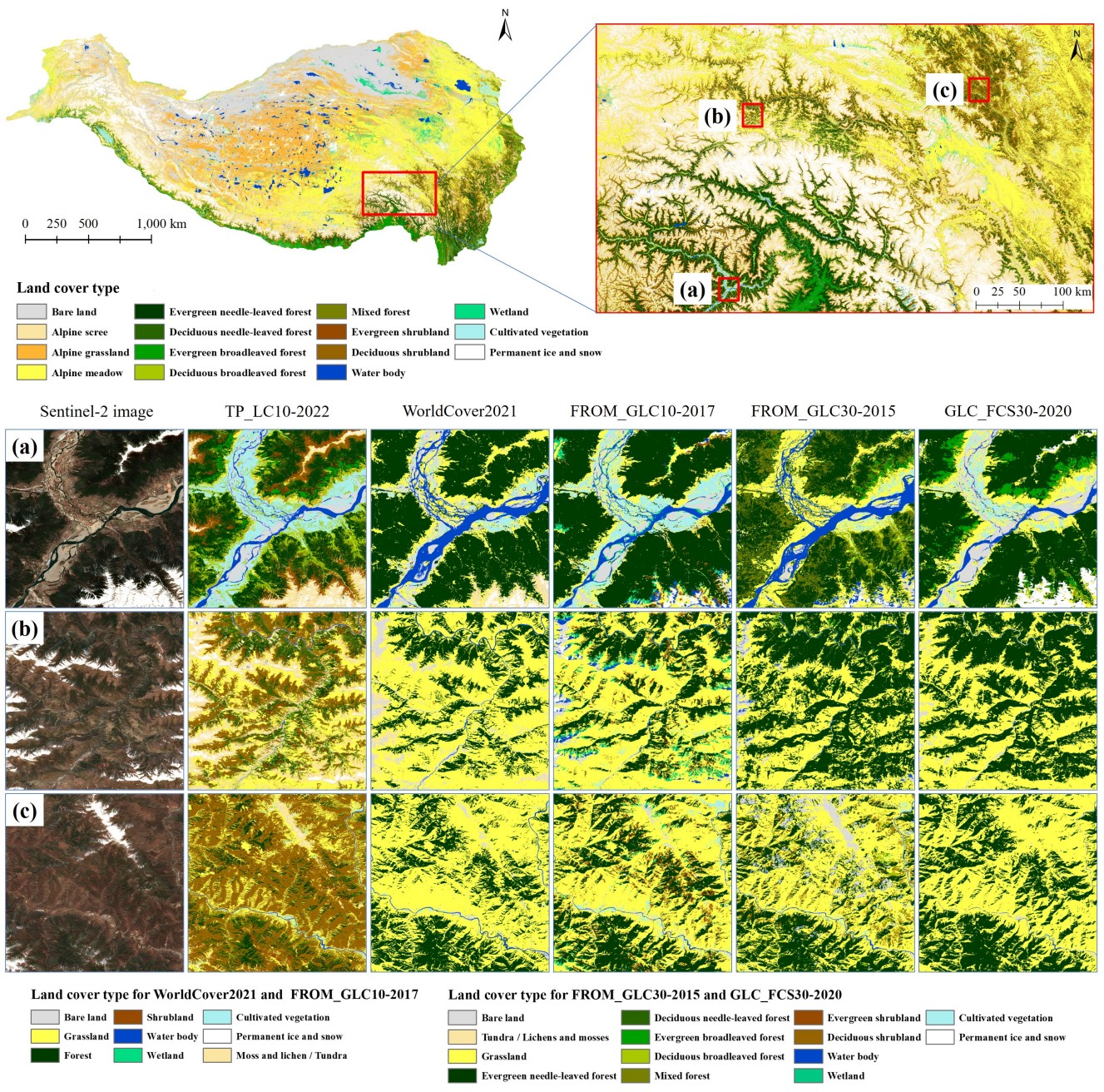

**Figure 7.** Comparison of TP_LC10-2022 with 4 other global land cover products in details. (a), (b) and (c) represent the $0.1° \times 0.1°$ grids for detailed comparisons. Legend fusion rules for WorldCover2021 and FROM_GLC10-2017 are provided in Table A2, and for FROM_GLC30-2015 and GLC_FCS30-2020, refer to Table A3.

**Table 5.** Comparison of mapping accuracy based on validation samples merged into 8 land cover types. Legend for the fusion rules of the 5 land cover products is provided in Table A2. Bold font denotes highest U.A. and P.A. within each land cover type, as well as the highest O.A. and Kappa among the 5 products.

| | | BL | GL | FST | SHR | WB | WL | CV | PIS | O.A | Kappa |
|---|---|---|---|---|---|---|---|---|---|---|---|
| GLC_FCS30-2020 | P.A. | 0.726 | 0.757 | **0.956** | 0.026 | 0.965 | 0.167 | 0.516 | **1.000** | 0.691 | 0.604 |
| | U.A. | 0.769 | 0.538 | 0.709 | 0.216 | 0.965 | **1.000** | 0.797 | 0.903 | | |
| FROM_GLC30-2015 | P.A. | 0.902 | 0.518 | 0.936 | 0.063 | 0.981 | 0.048 | 0.440 | 0.931 | 0.663 | 0.578 |
| | U.A. | 0.531 | 0.524 | 0.758 | 0.593 | 0.737 | 0.375 | 0.688 | 0.979 | | |
| FROM_GLC10-2017 | P.A. | 0.961 | 0.493 | 0.953 | 0.058 | 0.982 | 0.085 | 0.824 | 0.971 | 0.683 | 0.604 |
| | U.A. | 0.520 | 0.474 | 0.820 | 0.563 | **0.974** | 0.188 | 0.773 | **1.000** | | |
| WorldCover2021 | P.A. | 0.936 | 0.604 | 0.935 | 0.000 | 0.991 | 0.521 | 0.802 | **1.000** | 0.706 | 0.631 |
| | U.A. | 0.569 | 0.509 | 0.787 | 0.000 | 0.863 | 0.974 | **1.000** | 0.990 | | |
| TP_LC10-2022 | P.A. | **0.970** | **0.928** | 0.935 | **0.881** | **1.000** | **0.556** | **0.923** | 0.961 | **0.919** | **0.900** |
| | U.A. | **0.912** | **0.872** | **0.962** | **0.889** | **0.974** | 0.889 | 0.857 | 0.980 | | |

BL: bare land; GL: grassland; FST: forest; SHR: shrubland; WB: water body; WL: wetland; CV: cultivated vegetation;
PIS: Permanent ice and snow

lands as forests, particularly FROM_GLC30-2015 and GLC_FCS30-2020, whereas TP_LC10-2022 accurately differentiated between forests and shrubs. In Fig. 7c, both FROM_GLC10-2017 and FROM_GLC30-2015 depicted scattered shrublands but lacked continuity. These 4 products overestimated grasslands and underestimated the extent of shrubland areas. This discrepancy may stem from the similar phenological characteristics between deciduous shrublands and meadows, posing difficulties in their sole distinction based on spectral features (Li et al., 2021b). However, TP_LC10-2022 integrates topographic and climatic factors as classification features, facilitating precise differentiation between shrublands and grasslands.

Lakes and glaciers are the sentinels of global climate change and constitute the foundation of the TP as a crucial water source for surrounding regions (Zhang et al., 2017; Zhang and Duan, 2021). Precisely extracting the boundaries of lakes and glaciers is imperative for quantitatively monitoring lake expansion and glacier melting, as well as understanding the dynamic relationship between them and precipitation (Zhao et al., 2022b; Tong et al., 2016; Zhang et al., 2021a). Our land cover data, samples, and mapping methodology can serve as a baseline support for these endeavors (Yan et al., 2020; Korzeniowska and Korup, 2017), which facilitates the effective utilization of available water resources and promotes the sustainable development of the economy and society in the Greater Tibetan Plateau area and downstream regions of rivers originating from the TP (Ding et al., 2019).

Alpine forests play a crucial role in carbon storage and sequestration, thereby enhancing ecosystem services in the TP (Lin et al., 2023; Wang et al., 2022b; Zhao et al., 2023a). Our study revealed that TP_LC10-2022 identified the smallest forested area (8.60%), while GLC_FCS30-2020 and FROM_GLC30-2015 classified the largest and second-largest areas of alpine forest, respectively (12.86% and 11.89%) (Table A4). Conversely, the area of shrubland exhibits nearly the opposite

trend (Table A4). Confusion also arises between alpine grassland and bare land, potentially leading to variations in carbon storage estimation within each vegetation type. These discrepancies could impact efforts related to forest resource protection and grassland management for animal husbandry (Li et al., 2020; Yu et al., 2022).

Alpine screes are extensively distributed across the TP, yet they are frequently disregarded from other products. Our product presents the initial description of alpine scree vegetation locations, which will contribute to environmental monitoring and biodiversity research in the periglacial zone of the TP (Li et al., 2014). Shrublands play a vital role as carbon sinks in ecosystems and hold substantial implications for biomass estimation and global carbon cycling (Ma et al., 2021; Nie et al., 2018). TP_LC10-2022 accurately predicts the spatial distribution of shrublands, which holds considerable importance in forecasting

the impact of future changes in the biomass and carbon cycle on global-scale ecosystems (Chang et al., 2022).

High-resolution and accurate land cover data encompassing diverse vegetation types are crucial for monitoring large-scale alpine vegetation dynamics (Wang et al., 2023a, 2022b, 2020). For instance, relying on land cover maps such as ESA WorldCover as the foundation to examine tree lines and vegetation lines in the TP may lead to the underestimation of tree lines due to misclassifications of grasslands and shrublands (Fig. 7) (Zou et al., 2023). Additionally, the vegetation line may also be under-

estimated because of the absence of alpine scree (Fig. 7). In our future work, we aim to leverage the Sentinel-2, Sentinel-1, and other multisource data to annually generate TP_LC10 products. This approach will facilitate alpine vegetation monitoring and change detection, thereby enriching our comprehension of the dynamic TP amidst intensifying global climate change (Wang et al., 2022a).

## 5   Conclusions

We present a detailed land cover map including 12 vegetation types and 3 non-vegetation types at 10 m spatial resolution of the year 2022 for the Tibet Plateau (TP_LC10-2022) by integrating multi-source data including Sentinel-1, Sentinel-2, SRTM, CHIRPS, and ERA5-Land and comparing 4 classification models via GEE. The TP_LC10-2022 achieved an overall accuracy of 86.5% and a Kappa coefficient of 0.854% using the RF model, which outperforms other classification models, including GTB, MD and SVM. The comparisons between TP_LC10-2022 and 4 widely used land cover products (GLC_FCS30-2020,

FROM_GLC30-2015, FROM_GLC10-2017, and WorldCover2021) demonstrated that TP_LC10-2022 has higher overall accuracy and reflects the local-scale variations of vegetation types along latitudes. In particular, TP_LC10-2022 incorporated unique land cover types like alpine scree, alpine grassland, and alpine meadow, which accounts for 54.23% of the total coverage. Moreover, it accurately depicted the distribution of shrubland that occupied 4.63% of the TP and was underestimated in the other products. The proposed vegetation classification system for the TP can serve as a foundation for land cover mapping

in this region and a reference approach for mapping shrubland globally. The developed TP_LC10-2022 product can facilitate monitoring vegetation changes and studying the response to climate change in the TP.

*Data availability.* The TP_LC-2022 product generated in this paper is available at https://doi.org/10.5281/zenodo.8214981 (Huang et al., 2023a). Across the entire Tibetan Plateau, the TP_LC-2022 product is grouped by 54 3° × 3° tiles in the GeoTIFF format (EPSG: 4326), which are named "TP_LC10-2022_E**N**.tif", where "E**N**" explains the longitude and latitude information of the upper left corner of each regional land cover map. The multi-source data used in this study, including Sentinel-2, can be directly accessed from Google Earth Engine.

The corresponding sample dataset, produced by manual interpretation and field trips, is available at https://doi.org/10.5281/zenodo.8227942 (Huang et al., 2023b). The classification map can be viewed through https://cold-classifier.users.earthengine.app/view/tplc10-2022.

# Appendix A

**Table A1.** Optimal parameters for Random Forest (RF), Gradient Tree Boosting (GTB), Minimum Distance (MD), and Support Vector Machine (SVM) in this study.

| Model | Optical parameters |
|-------|--------------------|
| RF    | numberOfTrees: 100 |
| GTB   | numberOfTrees: 75  |
| MD    | metric: 'mahalanobis' |
|       | kNearest: 1        |
| SVM   | decisionProcedure: 'Voting' |
|       | kernelType: 'RBF'  |
|       | gamma: 0.000005    |
|       | cost: 2000         |

**Table A2.** Cross-walking table between different land cover products.

| Target type | TP_LC2022 | WorldCover2021 | FROM_GLC10-2017 | FROM_GLC30-2015 | GLC_FCS30-2020 |
|---|---|---|---|---|---|
| Bare land | Bare land | Bare / sparse vegetation<br>Built-up | Bare land<br>Impervious area | Bareland<br>Impervious surface | Bare areas<br>Impervious surfaces<br>Consolidated bare areas<br>Unconsolidated bare areas |
| Grassland | Alpine grassland<br>Alpine meadow | Grassland | Grassland | Natural grassland<br>Grassland, leaf-off | Grassland<br>Sparse vegetation<br>Sparse herbaceous<br>Herbaceous cover |
| Forest | Evergreen broadleaved forest<br>Deciduous broadleaved forest<br>Evergreen needle-leaved forest<br>Deciduous needle-leaved forest<br>Mixed forest | Tree cover | Forest | Broadleaf, leaf-on<br>Broadleaf, leaf-off<br>Needleleaf, leaf-on<br>Needleleaf, leaf-off<br>Mixed leaf, leaf-on | Open evergreen broadleaved forest<br>Closed evergreen broadleaved forest<br>Open deciduous broadleaved forest<br>Closed deciduous broadleaved forest<br>Open evergreen needle-leaved forest<br>Closed evergreen needle-leaved forest<br>Open deciduous needle-leaved forest<br>Closed deciduous needle-leaved forest |
| Shrubland | Evergreen Shrubland<br>Deciduous Shrubland | Shrubland | Shrubland | Shrubland, leaf-on<br>Shrubland, leaf-off | Shrubland<br>Evergreen Shrubland<br>Deciduous Shrubland |
| Water body | Water body | Permanent water bodies | Water body | Water | Water body |
| Wetland | Wetland | Herbaceous wetland | Wetland | Marshland<br>Mudflat<br>Marshland, leaf-off | Wetlands |
| Cultivated vegetation | Cultivated vegetation | Cropland | Cropland | Rice paddy<br>Greenhouse<br>Orchard<br>Bare farmland<br>Other (Cropland) | Rainfed cropland<br>Tree or shrub cover (Orchard)<br>Irrigated cropland |
| Permanent ice and snow | Permanent ice and snow | Snow and ice | Snow and ice | Snow<br>Ice | Permanent ice and snow |
| Excluded | Alpine scree | Moss and lichen | Tundra | Herbaceous tundra | Lichens and mosses |

● This table includes only land cover types present within the study area.

**Table A3.** Cross-walking table between FROM_GLC30-2015 and GLC_FCS30-2020.

| Target type | FROM_GLC30-2015 | GLC_FCS30-2020 |
|---|---|---|
| Bare land | Bareland<br>Impervious surface | Bare areas<br>Impervious surfaces<br>Consolidated bare areas<br>Unconsolidated bare areas |
| Grassland | Natural grassland<br>Grassland, leaf-off | Grassland<br>Sparse vegetation<br>Sparse herbaceous<br>Herbaceous cover |
| Evergreen broadleaved forest | Broadleaf, leaf-on | Open evergreen broadleaved forest<br>Closed evergreen broadleaved forest |
| Deciduous broadleaved forest | Broadleaf, leaf-off | Open deciduous broadleaved forest<br>Closed deciduous broadleaved forest |
| Evergreen needle-leaved forest | Needleleaf, leaf-on | Open evergreen needle-leaved forest<br>Closed evergreen needle-leaved forest |
| Deciduous needle-leaved forest | Needleleaf, leaf-off | Open deciduous needle-leaved forest<br>Closed deciduous needle-leaved forest |
| Mixed forest | Mixed leaf, leaf-on | |
| Evergreen shrubland | Shrubland, leaf-on | Evergreen Shrubland |
| Deciduous shrubland | Shrubland, leaf-off | Deciduous Shrubland |
| Water body | Water | Water body |
| Wetland | Marshland<br>Mudflat<br>Marshland, leaf-off | Wetlands |
| Cultivated vegetation | Rice paddy<br>Greenhouse<br>Orchard<br>Bare farmland<br>Other (Cropland) | Rainfed cropland<br>Tree or shrub cover (Orchard)<br>Irrigated cropland |
| Permanent ice and snow | Snow<br>Ice | Permanent ice and snow |
| Tundra / Lichens and mosses | Herbaceous tundra | Lichens and mosses |

● This table includes only land cover types present within the study area.

● The 'cloud' class in the FROM_GLC30-2015 and 'shrubland' class in the GLC_FCS30-2020 have been omitted from the table due to their small area.

**Table A4.** Area statistical results for land cover products in the Tibetan Plateau.

| Land Cover type | TP_LC10-2022 Area | Proportion | FROM_GLC30-2015 Area | Proportion | GLC_FCS30-2020 Area | Proportion | WorldCover2021 Area | Proportion | FROM_GLC10-2017 Area | Proportion |
|---|---|---|---|---|---|---|---|---|---|---|
| BL | 58.75 | 19.05% | 147.67 | 47.89% | 45.71 | 14.82% | 134.75 | 43.70% | 156.45 | 50.74% |
| AG | 50.83 | 16.48% | 96.75 | 31.38% | 189.44 | 61.44% | 108.44 | 35.17% | 89.35 | 28.98% |
| AM | 73.25 | 23.76% | | | | | | | | |
| ENF | 11.44 | 3.71% | 27.91 | 9.05% | 31.52 | 10.22% | 28.49 | 9.24% | 29.46 | 9.55% |
| DNF | 2.26 | 0.73% | 0.02 | 0.01% | 0.37 | 0.12% | | | | |
| EBF | 4.53 | 1.47% | 2.94 | 0.95% | 3.45 | 1.12% | | | | |
| DBF | 6.80 | 2.20% | 1.69 | 0.55% | 4.31 | 1.40% | | | | |
| MF | 1.46 | 0.47% | 4.10 | 1.33% | 0.00 | 0.00% | | | | |
| ES | 3.28 | 1.06% | 1.70 | 0.55% | 0.22 | 0.07% | 0.37 | 0.12% | 1.59 | 0.51% |
| DS | 11.02 | 3.57% | 0.41 | 0.13% | 4.13 | 1.34% | | | | |
| WB | 6.43 | 2.09% | 12.38 | 4.02% | 6.05 | 1.96% | 6.86 | 2.22% | 10.06 | 3.26% |
| WL | 6.84 | 2.22% | 0.19 | 0.06% | 0.55 | 0.18% | 0.37 | 0.12% | 2.06 | 0.67% |
| CV | 5.14 | 1.67% | 2.04 | 0.66% | 2.81 | 0.91% | 1.35 | 0.44% | 3.02 | 0.98% |
| PIS | 23.18 | 7.52% | 10.46 | 3.39% | 19.08 | 6.19% | 12.95 | 4.20% | 16.36 | 5.30% |
| AS / Tundra / Lichen / Moss | 43.15 | 13.99% | 0.05 | 0.01% | 0.00 | 0.00% | 14.77 | 4.79% | 0.00 | 0.00% |
| Total | 308.34 | 100.00% | 308.31 | 99.99% | 307.66 | 99.78% | 308.34 | 100.00% | 308.34 | 100.00% |

BL: bare land; AG: alpine grassland; AM: alpine meadow; ENF: evergreen needle-leaved forest; DNF: deciduous needle-leaved forest;

EBF: evergreen broadleaved forest; DBF: deciduous broadleaved forest; MF: mixed forest; ES: evergreen shrubland; DS: deciduous shrubland;

WB: water body; WL: wetland; CV: cultivated vegetation; PIS: permanent ice and snow; AS: alpine scree

● The unit of area is ten thousand square kilometers, and the unit of proportion is percent.

● Please refer to Table A3 for the merging rules of land cover for FROM_GLC30-2015 and GLC_FCS30-2020.

● The 'cloud' class in the FROM_GLC30-2015 and 'shrubland' class in the GLC_FCS30-2020 product have been omitted from the table due to their small area.

● All built-up pixels are merged with bare land.

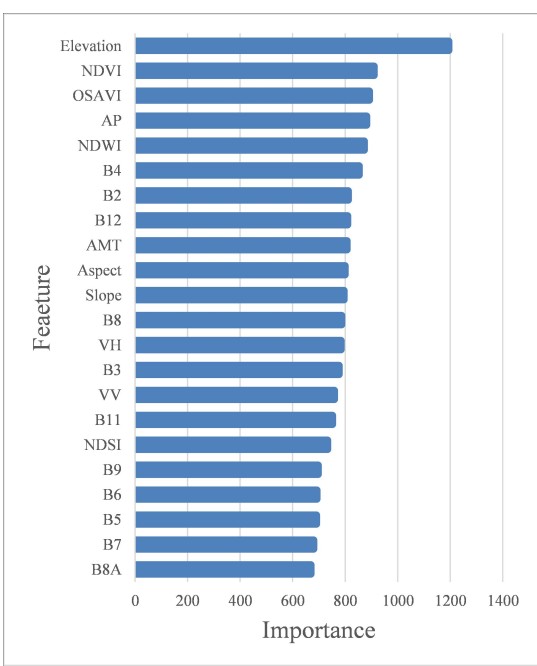

**Figure A1.** Statistical chart of the importance of different features for the Random Forest classification model. AP: annual precipitation; AMT: annual mean temperature.

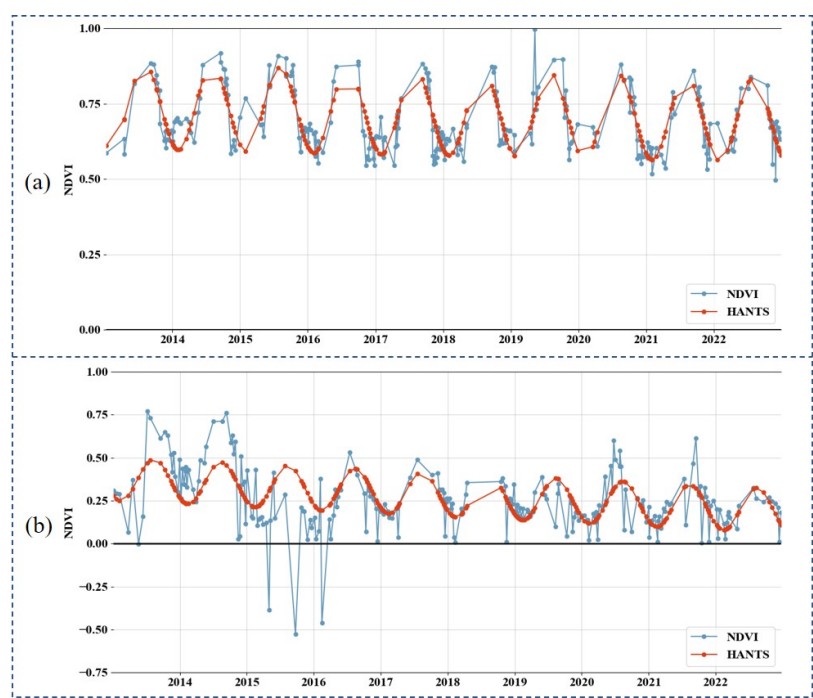

**Figure A2.** Landsat NDVI time series and HANTS filtered NDVI time series for stability verification. (a) depicts a deciduous needle-leaved forest, while (b) shows a transition from forest to farmland at the edge of the deciduous broadleaved forest in 2015, where it was annually cultivated following deforestation.

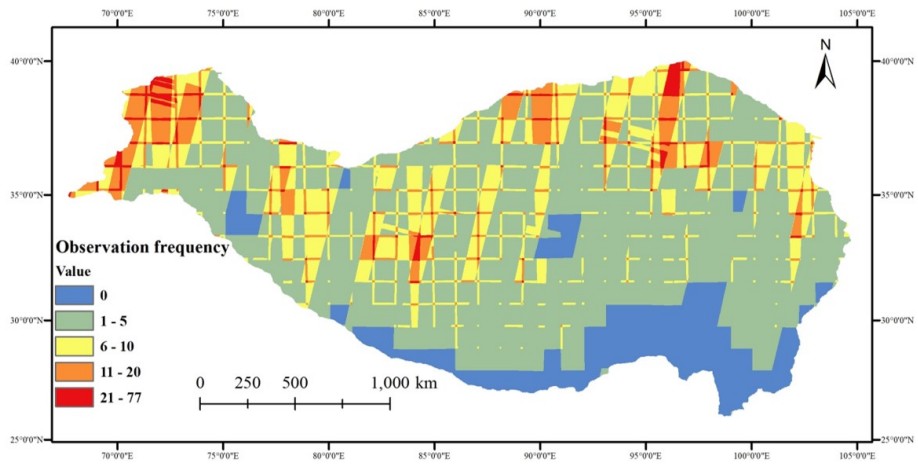

**Figure A3.** Number of available observations for the Sentinel-2 optical data in the Tibetan Plateau during summer in 2022 (June 1, 2022, to August 31, 2022) with cloud cover < 10%.

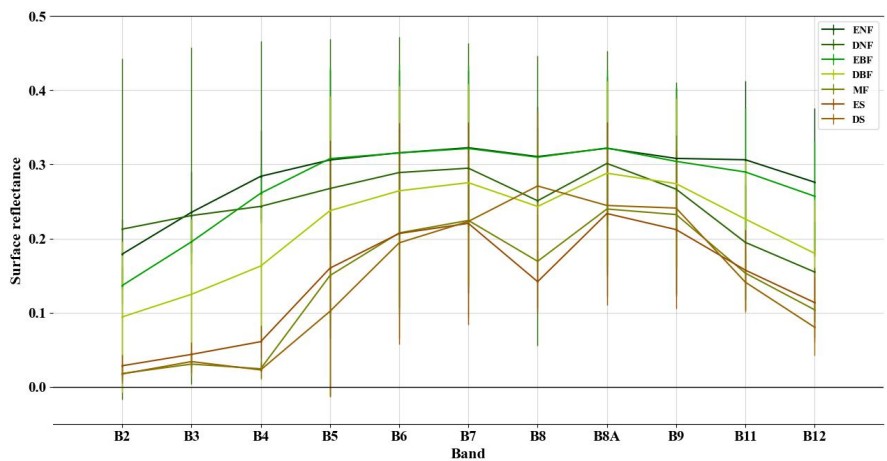

**Figure A4.** Sentinel-2 spectral curves for forest and shrubland types. The spectral curve for each type was derived by calculating the average and standard deviation of surface reflectance across all samples for the processed cloud-free Sentinel-2 median composite for 2022 in the Tibetan Plateau. ENF: evergreen needle-leaved forest; DNF: deciduous needle-leaved forest; EBF: evergreen broadleaved forest; DBF: deciduous broadleaved forest; MF: mixed forest; ES: evergreen shrubland; DS: deciduous shrubland.

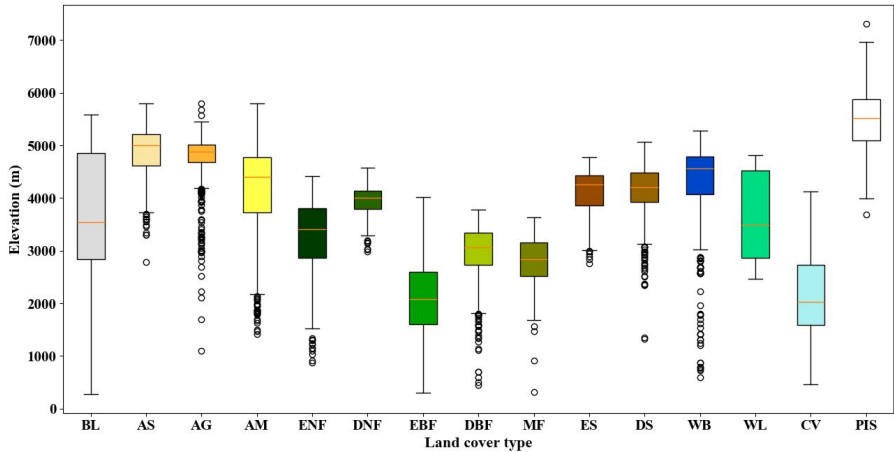

**Figure A5.** Box plot derived from SRTM for the distribution of sample elevation across different land cover types in the Tibetan Plateau. BL: bare land; AS: alpine scree; AG: alpine grassland; AM: alpine meadow; ENF: evergreen needle-leaved forest; DNF: deciduous needle-leaved forest; EBF: evergreen broadleaved forest; DBF: deciduous broadleaved forest; MF: mixed forest; ES: evergreen shrubland; DS: deciduous shrubland; WB: water body; WL: wetland; CV: cultivated vegetation; PIS: permanent ice and snow.

*Author contributions.* FT conceived the study. XH and YY collected the samples and carried out the analysis while LF provided the technical support. XH and YY wrote the original draft. FT and JL carried out the field trip. All authors helped revise the draft.

*Competing interests.* The authors declare that they have no conflict of interest.

*Disclaimer.* TEXT

*Acknowledgements.* This work is funded by the National Key Research and Development Program of China (2020YFA0608704), the National Natural Science Foundation of China (Grant No. 42001299), and the Seed Fund Program for Sino-Foreign Joint Scientific Research Platform of Wuhan University (No. WHUZZJJ202205).

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
