# Peer review of "A 10 m resolution land cover map of the Tibetan Plateau with detailed vegetation types"

_Earth System Science Data, 2023_

## Author Comment (AC1)

**A 10 m resolution land cover map of the Tibetan Plateau with detailed vegetation types**

Anonymous Referee #2

**General comments:**

**This manuscript 'A 10 m resolution land cover map of the Tibetan Plateau with detailed vegetation types' produced a 10 m resolution TP land cover map to address the issue of low spatial resolution and incomplete vegetation type coverage in the existing TP land cover dataset. The generated TP_LC10-2022 product will be a valuable data for the study of this region, but the method employed in the manuscript lacks innovation. In addition, there are still many areas that need improvement.**

**Response:** We are grateful for your kind acknowledgment of the value of our dataset and thank you for providing insightful comments and detailed suggestions. Following your constructive feedback, we have revised the text to strengthen the clarity and accuracy of this manuscript.

It is very true that the classification methods used in this study are widely used but also well recognized in the research community. We believe that in the current stage, with the rapid advance of machine learning algorithms and computing power, the size, accuracy, and separability of the training samples are more important than the classification algorithms employed. Therefore, in this study, we emphasize more on bridging the gaps between existing land cover products and the requirements of ecologically meaningful applications in TP, that is, the new classification system. Yet, we also tested several machine-learning algorithms and selected the best one to achieve high classification accuracy.

**Specific comments:**

1. **First, it is recommended to separate the data and methods sections. The current structure is somewhat confusing. It is suggested to merge sections 2.1 to 2.2 under the main heading "2. Study Area and Data." Also, starting from section 2.3 to 2.3.5, it is suggested to be included in Section 3 as "3. Methodology."**

**Reply 1:**

Thank you for the insightful suggestion. We have incorporated your feedback by consolidating sections 2.1 to 2.2 into the main heading "2. Study Area and Data." Furthermore, sections 2.3 to 2.3.5 have been integrated into section 3, resulting in a clearer and more cohesive structure. Please refer to our revised manuscript.

**2. Secondly, in section 2.3.1, why did the authors state "The advantages of our classification system are as follows"? Here, the authors should introduce the basis for constructing this classification system, rather than directly discussing its advantages. Moreover, the content of this section seems more like it is introducing the basis for constructing the classification system. Moreover, what exactly is the content "Discriminability in Remote Sensing Imagery" trying to state? Is this part related to the classification system? Furthermore, didn't the authors developed their product using Sentinel data? How come 0.3m of Google Earth imagery was involved here?**

**Reply 2:**

Thanks for this detailed and constructive comment. We agree that section 2.3.1 is a description of the basis of our classification system. We have revised the text in this section accordingly (Lines 116-117).

The section titled "Discriminability in Remote Sensing Imagery" did relate to the classification system. To make it clearer, we have rephrased the sentence to "Discriminability of different vegetation functional types in remote sensing imagery". Here, we aimed to demonstrate the effectiveness of our classification system during the image interpretation stage. Google Earth imagery serves as a valuable resource for distinguishing various land cover types due to its high resolution. Obtaining a substantial number of land cover samples, particularly in remote areas of the TP, would be nearly impossible without access to such high-resolution remote sensing images.

It is important to emphasize the significance of Sentinel-2 data, particularly in discriminating land cover types during the image classification stage. The multispectral bands provided by Sentinel-2 are crucial for accurately classifying diverse land cover types within the TP. To address this concern, we have incorporated additional information regarding Sentinel-2 data in the revised text (Lines 126-131), which reads:

*During the classification stage, we can effectively differentiate various land cover types, including diverse vegetation, utilizing the discriminative capabilities of the multispectral bands of Sentinel-2 (Liu et al., 2023). Moreover, the incorporation of high-resolution Google Earth imagery, with a spatial resolution of up to 0.3 meters, enhances the discernibility of land cover types during the sample selection phase. This ensures the feasibility of visually interpreting large-scale samples from remote sensing imagery and obtaining reliable and up-to-date information (Gong et al., 2013).*

**Reference:**

Gong, P., Wang, J., Yu, L., Zhao, Y., Zhao, Y., Liang, L., Niu, Z., Huang, X., Fu, H., Liu, S., Li, C., Li, X., Fu, W., Liu, C., Xu, Y., Wang, X., Cheng, Q., Hu, L., Yao, W., … Chen, J. (2013). Finer resolution observation and monitoring of global land cover: First

mapping results with Landsat TM and ETM+ data. International Journal of Remote Sensing, 34(7), 2607–2654. https://doi.org/10.1080/01431161.2012.748992

Liu, X., Frey, J., Munteanu, C., Still, N., & Koch, B. (2023). Mapping tree species diversity in temperate montane forests using Sentinel-1 and Sentinel-2 imagery and topography data. Remote Sensing of Environment, 292, 113576. https://doi.org/10.1016/j.rse.2023.113576

**3. Thirdly, why were the training and validation sets configured as 4:1? Typically, they are set to 7:3.**

**Reply 3:**

Considering that we derived 22 features for the classification and the land cover in the TP is highly heterogeneous, the machine learning model needs to learn the complex distribution of different data features. Compared to the case of a 7:3 ratio, 80% of the data is used to train the model, which allows the model to better learn the features and patterns of the data and improve the model's fitting ability (Ramezan et al., 2021). We have added an explanation of the ratio of training and validation sets (Lines 177-179), which reads:

*We adjusted the ratio of training to validation samples to 4:1 instead of the commonly used 7:3 to enhance the model's fitting capability to handle the complex distribution of features (Ramezan et al., 2021).*


[Figure]

*Figure A4. Sentinel-2 spectral curves for forest and shrubland types. The spectral curve for each type was derived by calculating the average and standard deviation of surface reflectance across all samples for the processed cloud-free Sentinel-2 median composite for 2022 in the Tibetan Plateau. ENF: evergreen needle-leaved forest; DNF: deciduous needle-leaved forest; EBF: evergreen broadleaved forest; DBF: deciduous broadleaved forest; MF: mixed forest; ES: evergreen shrubland; DS: deciduous shrubland.*

[Figure]

*Figure A5. Box plot derived from SRTM for the distribution of sample elevation across different land cover types in the Tibetan Plateau. BL: bare land; AS: alpine scree; AG: alpine grassland; AM: alpine meadow; ENF: evergreen needle-leaved forest; DNF: deciduous needle-leaved forest; EBF: evergreen broadleaved forest; DBF: deciduous broadleaved forest; MF: mixed forest; ES: evergreen shrubland; DS: deciduous shrubland; WB: water body; WL: wetland; CV: cultivated vegetation; PIS: permanent ice and snow.*

**Reply 6:**

Indeed, the classification methods we used to generate the land cover map are widely used but also well recognized in the research community. We believe that in the current stage, with the rapid advance of machine learning algorithms and computing power, the size, accuracy, and separability of the training samples are more important

than the classification algorithms employed. Therefore, in this study, we emphasize more on bridging the gaps between existing land cover products and the requirements of ecologically meaningful applications in TP, that is, the new classification system. Yet, we also tested several machine-learning algorithms and selected the best one to achieve high classification accuracy. Furthermore, our methodology is meticulously crafted, comprehensive, and resilient.

Land cover mapping research holds paramount importance in Earth science, and our datasets, encompassing the land cover map and samples for the TP, can serve as a robust foundation for further research endeavors in this pivotal region. For this, we strengthened our discussion in terms of its implications for the sustainable use of available resources in practice, for policy making, and for further research (Lines 312-340), which reads:

**1.  For sustainable use of available resources in practice:**

[revised manuscript text omitted]

Li, J., Gong, J., Guldmann, J.-M., Li, S., & Zhu, J. (2020). Carbon Dynamics in the Northeastern Qinghai–Tibetan Plateau from 1990 to 2030 Using Landsat Land Use/Cover Change Data. Remote Sensing, 12(3), 528. https://doi.org/10.3390/rs12030528

Li, X.-H., Zhu, X.-X., Niu, Y., & Sun, H. (2014). Phylogenetic clustering and overdispersion for alpine plants along elevational gradient in the Hengduan Mountains Region, southwest China: Phylogenetic structure along elevational gradient. Journal of Systematics and Evolution, 52(3), 280–288. https://doi.org/10.1111/jse.12027

Lin, Y., Xiao, J.-T., Kou, Y.-P., Zu, J.-X., Yu, X.-R., & Li, Y.-Y. (2023). Aboveground carbon sequestration rate in alpine forests on the eastern Tibetan Plateau: Impacts of future forest management options. Journal of Plant Ecology, 16(3), rtad001. https://doi.org/10.1093/jpe/rtad001

Ma, H., Mo, L., Crowther, T. W., Maynard, D. S., van den Hoogen, J., Stocker, B. D., Terrer, C., & Zohner, C. M. (2021). The global distribution and environmental drivers of aboveground versus belowground plant biomass. Nature Ecology & Evolution, 5(8), 1110–1122. https://doi.org/10.1038/s41559-021-01485-1

Nie, X., Yang, L., Xiong, F., Li, C., Fan, L., & Zhou, G. (2018). Aboveground biomass of the alpine shrub ecosystems in Three-River Source Region of the Tibetan Plateau. Journal of Mountain Science, 15(2), 357–363. https://doi.org/10.1007/s11629-016-4337-0

Tong, K., Su, F., & Xu, B. (2016). Quantifying the contribution of glacier meltwater in the expansion of the largest lake in Tibet. Journal of Geophysical Research: Atmospheres, 121(19). https://doi.org/10.1002/2016JD025424

Wang, F., Ma, Y., Darvishzadeh, R., & Han, C. (2023). Annual and Seasonal Trends of Vegetation Responses and Feedback to Temperature on the Tibetan Plateau since the 1980s. Remote Sensing, 15(9), 2475. https://doi.org/10.3390/rs15092475

Wang, Y., Li, D., Ren, P., Ram Sigdel, S., & Camarero, J. J. (2022). Heterogeneous Responses of Alpine Treelines to Climate Warming across the Tibetan Plateau. Forests, 13(5), 788. https://doi.org/10.3390/f13050788

Wang, Z., Song, W., & Yin, L. (2022). Responses in ecosystem services to projected land cover changes on the Tibetan Plateau. Ecological Indicators, 142, 109228. https://doi.org/10.1016/j.ecolind.2022.109228

Wang, Z., Wu, J., Niu, B., He, Y., Zu, J., Li, M., & Zhang, X. (2020). Vegetation Expansion on the Tibetan Plateau and Its Relationship with Climate Change. Remote Sensing, 12(24), 4150. https://doi.org/10.3390/rs12244150

Yan, D., Huang, C., Ma, N., & Zhang, Y. (2020). Improved Landsat-Based Water and Snow Indices for Extracting Lake and Snow Cover/Glacier in the Tibetan Plateau. Water, 12(5), 1339. https://doi.org/10.3390/w12051339

Yu, C., Xu, L., Li, M., & He, N. (2022). Phosphorus storage and allocation in vegetation on the Tibetan Plateau. Ecological Indicators, 145, 109636. https://doi.org/10.1016/j.ecolind.2022.109636

Zhang, G., & Duan, S. (2021). Lakes as sentinels of climate change on the Tibetan Plateau. All Earth, 33(1), 161–165. https://doi.org/10.1080/27669645.2021.2015870

Zhang, G., Yao, T., Piao, S., Bolch, T., Xie, H., Chen, D., Gao, Y., O'Reilly, C. M., Shum, C. K., Yang, K., Yi, S., Lei, Y., Wang, W., He, Y., Shang, K., Yang, X., & Zhang, H. (2017). Extensive and drastically different alpine lake changes on Asia's high plateaus during the past four decades. Geophysical Research Letters, 44(1), 252–260. https://doi.org/10.1002/2016GL072033

Zhang, J., Hu, Q., Li, Y., Li, H., & Li, J. (2021). Area, lake-level and volume variations of typical lakes on the Tibetan Plateau and their response to climate change, 1972–2019. Geo-Spatial Information Science, 24(3), 458–473. https://doi.org/10.1080/10095020.2021.1940318

Zhao, H., Guo, B., & Wang, G. (2023). Spatial–Temporal Changes and Prediction of Carbon Storage in the Tibetan Plateau Based on PLUS-InVEST Model. Forests, 14(7), 1352. https://doi.org/10.3390/f14071352

Zhao, R., Fu, P., Zhou, Y., Xiao, X., Grebby, S., Zhang, G., & Dong, J. (2022). Annual 30-m big Lake Maps of the Tibetan Plateau in 1991–2018. Scientific Data, 9(1), 164.

https://doi.org/10.1038/s41597-022-01275-9

Zou, L., Tian, F., Liang, T., Eklundh, L., Tong, X., Tagesson, T., Dou, Y., He, T., Liang, S., & Fensholt, R. (2023). Assessing the upper elevational limits of vegetation growth in global high-mountains. Remote Sensing of Environment, 286, 113423. https://doi.org/10.1016/j.rse.2022.113423

---

## Author Comment (AC2)

**A 10 m resolution land cover map of the Tibetan Plateau with detailed vegetation types**

Anonymous Referee #1

**General comments:**

**The authors introduced a new classification system and produced a detailed land cover map of the Tibet Plateau (TP) area in 2022, which is significant for climate change studies. The method and results are well-presented. However, there are some questions or issues.**

**Response:** We are grateful for your kind acknowledgment of the value of our dataset and thank you for providing insightful comments and detailed suggestions. Following your constructive feedback, we have revised the text to strengthen the clarity and accuracy of this manuscript.

**Specific comments:**

1. **Lines 242–245 mentioned that the reason for not using time series is the dense cloud cover in southeastern TP. Could you provide a quantification of the cloud coverage in this region?**

**Reply 1:**

Thank you for pointing this out. Lowering the threshold of cloud filtering results in the reduction of image pixels available for analysis, particularly in the southeastern TP, where has heavy cloud contamination (Tang et al., 2022). Conversely, raising this threshold to a higher level compromises the quality control of Sentinel-2 images while maintaining image integrity. We have added a quantification of filtered Sentinel-2 imagery in the TP in the revised manuscript (Lines 273-278), which reads:

*For example, during the summer of 2022 (June-August), when setting the "CLOUDY_PIXEL_PERCENTAGE" parameter to 10%, 20%, 30%, and 40%, and applying QA band masking, we lost 13.59%, 5.81%, 2.44%, and 1.32% of the Sentinel-2 image area in the TP. The removed pixels are concentrated mainly in the cloudy southeastern TP (only shown for 10% threshold in Fig. A3) (Tang et al., 2022). This constraint can preclude the attainment of desired outcomes in regions where cloud-free image availability is low (Chu et al., 2021; Coluzzi et al., 2018).*

[Figure]

*Figure A3. Number of available observations for the Sentinel-2 optical data in the Tibetan Plateau during summer in 2022 (June 1, 2022, to August 31, 2022) with cloud cover <10%.*

**Reference:**

Chu, D., Shen, H., Guan, X., Chen, J. M., Li, X., Li, J., & Zhang, L. (2021). Long time-series NDVI reconstruction in cloud-prone regions via spatio-temporal tensor completion. Remote Sensing of Environment, 264, 112632. https://doi.org/10.1016/j.rse.2021.112632

Coluzzi, R., Imbrenda, V., Lanfredi, M., & Simoniello, T. (2018). A first assessment of the Sentinel-2 Level 1-C cloud mask product to support informed surface analyses. Remote Sensing of Environment, 217, 426–443.

Tang, J., Guo, X., Chang, Y., Lu, G., & Qi, P. (2022). Long-term variations of clouds and precipitation on the Tibetan Plateau and its subregions, and the associated mechanisms. International Journal of Climatology, 42(16), 9003–9022. https://doi.org/10.1002/joc.7792

2. **Lines 131–141. The Landsat NDVI time series from 2013 to 2022 was used to assist in selecting samples. I also noted that the study selected dense samples in the southeastern TP. Could the cloud cover in southeastern TP affect the Landsat time series from 2013 to 2022 and subsequently impact the accuracy of the sample selection?**

**Reply 2:**

The dense cloud cover does not affect the accuracy of sample selection over the entire study area. However, cloud cover does reduce the number of available NDVI observations from Landsat images in the southeastern TP. To mitigate this impact on the cloudy regions, we implemented the following two steps:

- Filtering out pixels with cloud coverage greater than 50% to eliminate severely contaminated pixels.

- Using harmonic analysis of time series (HANTS) model for data interpolation and smoothing to remove outliers and reconstruct missing data.

As a result of this process, only a small fraction of pixels remained severely contaminated, which were excluded from our sample selection.

**3. In Fig. 3, the NDVI time series for evergreen needle-leaved forest, evergreen broadleaved forest, and evergreen shrubland look very similar. Can the NDVI time series effectively distinguish between these land cover types?**

**Reply 3:**

Thank you for pointing this out. It is difficult to distinguish certain land cover types only using the Landsat NDVI time series, such as evergreen needle-leaved forest, evergreen broadleaved forest, and evergreen shrubland, due to their similar vegetation characteristics. However, the decision was made not only based on NDVI but also on their distinct crown shapes and texture characteristics that are visible in Google Earth images (see Fig. 3 and the figure below). We have revised the text to make it clearer to understand (Lines 164-166), which reads:

*Evergreen needle-leaved forests, evergreen broadleaved forests, and evergreen shrublands exhibit similar trends and values in NDVI time series. However, they can be discerned in Google Earth images based on their distinctive crown shapes and textures (Fig. 3).*

[Figure]

*Figure R1 (only for response). Selected examples of auxiliary data derived from Google Earth imagery, including evergreen needle-leaved forest, evergreen broadleaved forest, evergreen shrubland, and transition zone between evergreen needle-leaved forest (left half) and evergreen shrubland (right half).*

**4. What is the proportion of samples that are directly visually interpreted from Google Earth images, samples using NDVI time series as auxiliary, and samples using only NDVI time series without Google Earth images?**

**Reply 4:**

All samples were interpreted based on Google Earth images, with subsequent verification using NDVI time series as a supplementary measure to ensure stability and detect phenology. No samples are selected using only NDVI time series in our study. We apologize for the unclarity and have revised the text to make it clearer to understand (Lines 158-160), which reads:

*All samples were interpreted based on Google Earth images, with subsequent verification using NDVI time series as a supplementary measure to ensure stability and detect phenology.*

**5. How can you eliminate the impact of land cover changes that may have occurred between 2013 and 2022 on the Landsat time series used in the sample selection?**

**Reply 5:**

As described in Lines 148-150, we utilized Landsat time series to identify changes in land cover, including deforestation, during our stability verification process. The figures provided below illustrate examples of this verification process. Fig. A2(b) depicts a typical NDVI time series of deciduous needle-leaved forest derived from Landsat 7 and Landsat 8, while Fig. A2(a) shows a site where deforestation occurred. Through the stability verification process, sample sites that changed in selected years (2013-2022) are excluded. We have clarified this in our revised manuscript (Lines 155-156), which reads:

*By following the steps outlined above, we detected land cover changes during 2013-2022 using Landsat NDVI time series (Fig. A2). This approach helps to avoid selecting sites where land cover change has occurred.*

[Figure]

*Figure A2. Landsat NDVI time series and HANTS filtered NDVI time series for stability verification. (a) depicts a deciduous needle-leaved forest, (b) shows a transition from forest to farmland at the edge of the deciduous broadleaved forest in 2015, where it was annually cultivated following deforestation.*

**6. According to Table A2, impervious surfaces or built-up areas are considered as bareland in the classification system. However, I noticed that built-up areas in cities such as Xining and Lhasa are incorrectly classified as cultivated vegetation and other land cover types in your product. I also noticed that the barelands in your training samples do not seem to include built-up area samples.**

**Reply 6:**

Thank you for pointing out the misclassification of built-up areas in some cities as other land cover types in our product. We have added descriptions about why we merged built-up areas and bare land in our classification system (Lines 132-136), which reads:

*In this study, we did not specifically select samples of built-up areas and instead categorized bare land together with built-up areas for two primary reasons. Firstly, built-up areas account for only 0.092% of the total area in ESA WorldCover2021, highlighting their relatively small extent compared to other land cover types (Zanaga et al., 2022). Secondly, bare land in our product exhibits spectral characteristics similar to those of built-up areas, resulting in the classification of most built-up areas as bare land (H. Li et al., 2017).*

Furthermore, we discussed this issue in our revised manuscript (Lines 266-268), which reads:

*In addition , the spectral variations within urban areas have also resulted in substantial uncertainties. Our approach of categorizing built-up areas and bare land may lead to misclassification of urban pixels. To minimize the uncertainties in urban areas on our final map, we applied the ESRI land cover map in 2022 to mask off urban pixels (Karra et al., 2021).*

**Reference:**

Karra, K., Kontgis, C., Statman-Weil, Z., Mazzariello, J. C., Mathis, M., & Brumby, S. P. (2021). Global land use / land cover with Sentinel 2 and deep learning. 2021 IEEE International Geoscience and Remote Sensing Symposium IGARSS, 4704–4707. https://doi.org/10.1109/IGARSS47720.2021.9553499

Li, H., Wang, C., Zhong, C., Su, A., Xiong, C., Wang, J., & Liu, J. (2017). Mapping Urban Bare Land Automatically from Landsat Imagery with a Simple Index. Remote Sensing, 9(3), 249. https://doi.org/10.3390/rs9030249

Zanaga, D., Van De Kerchove, R., Daems, D., De Keersmaecker, W., Brockmann, C., Kirches, G., Wevers, J., Cartus, O., Santoro, M., Fritz, S., Lesiv, M., Herold, M., Tsendbazar, N.-E., Xu, P., Ramoino, F., and Arino, O.: ESA WorldCover 10 m 2021 v200, Zenodo [data set], https://doi.org/10.5281/zenodo.5571936, 2022

**7. Lines 159–161. "Interannual" refers to two or more years, but you have only selected images from one year. Did you mean "Annual"?**

**Reply 7:**

Thank you for pointing out this mistake. We have rectified it in our revised manuscript (Line 78).

**8. You used almost all bands from Sentinel-2 with four additional indices. The information provided by some of the bands may be duplicated. For example, the wavelengths of B8 and B8A are close. Is it sufficient to use only one of them?**

**Reply 8:**

Thank you for this comment! Using either B8 or B8A alone is adequate to achieve high overall accuracy. However, incorporating B8A brings a minor improvement in the overall accuracy, despite its relatively lower importance in the Random Forest model used in this study. For instance, employing only the B8 band yields an overall accuracy of 86.1%, while incorporating B8A improves it to 86.5%. We have added the explanation of band selection in our revised manuscript (Lines 181-186), which reads:

*The selected input bands for Sentinel-2 included B2-B8, B8A, B9, B11, and B12. Among these bands, B2-B8, B11, and B12 have been demonstrated to be effective in*

*classifying deciduous and coniferous tree species (Immitzer et al., 2016; C. Li et al., 2021). Additionally, B8A is suitable for boreal landscape classification (Abdi, 2020), while B9 values demonstrate differences between bare soil and vegetation-covered areas (Zhao et al., 2023), making them useful for classification purposes. For Sentinel-1 images, utilizing both VV and VH can enhance classification accuracy, leading to their selection as input features (Jacob et al., 2020; Steinhausen et al., 2018).*

**Reference:**

Abdi, A. M. (2020). Land cover and land use classification performance of machine learning algorithms in a boreal landscape using Sentinel-2 data. Giscience & Remote Sensing, 57(1), 1–20. https://doi.org/10.1080/15481603.2019.1650447

Immitzer, M., Vuolo, F., & Atzberger, C. (2016). First Experience with Sentinel-2 Data for Crop and Tree Species Classifications in Central Europe. Remote Sensing, 8(3), 166. https://doi.org/10.3390/rs8030166

Jacob, A. W., Vicente-Guijalba, F., Lopez-Martinez, C., Lopez-Sanchez, J. M., Litzinger, M., Kristen, H., Mestre-Quereda, A., Ziolkowski, D., Lavalle, M., Notarnicola, C., Suresh, G., Antropov, O., Ge, S., Praks, J., Ban, Y., Pottier, E., Mallorqui Franquet, J. J., Duro, J., & Engdahl, M. E. (2020). Sentinel-1 InSAR Coherence for Land Cover Mapping: A Comparison of Multiple Feature-Based Classifiers. IEEE Journal of Selected Topics in Applied Earth Observations and Remote Sensing, 13, 535–552. https://doi.org/10.1109/JSTARS.2019.2958847

Li, C., Ma, Z., Wang, L., Yu, W., Tan, D., Gao, B., Feng, Q., Guo, H., & Zhao, Y. (2021). Improving the Accuracy of Land Cover Mapping by Distributing Training Samples. Remote Sensing, 13(22), 4594. https://doi.org/10.3390/rs13224594

Steinhausen, M. J., Wagner, P. D., Narasimhan, B., & Waske, B. (2018). Combining Sentinel-1 and Sentinel-2 data for improved land use and land cover mapping of monsoon regions. International Journal of Applied Earth Observation and Geoinformation, 73, 595–604. https://doi.org/10.1016/j.jag.2018.08.011

Zhao, Y., Lei, S., Zhu, G., Shi, Y., Wang, C., Li, Y., Su, Z., & Wang, W. (2023). An Algorithm to Retrieve Precipitable Water Vapor from Sentinel-2 Data. Remote Sensing, 15(5), 1201. https://doi.org/10.3390/rs15051201

**9. In the comparison with other products, these products are from different years. Due to land cover changes, comparison across years will introduce some error. Using validation samples in 2022 is also unfair to products of other years. These issues need to be discussed.**

**Reply 9:**

Thank you. Our samples remain stable from 2013 to 2022, as explained in Reply 5. These consistent validation samples can be utilized to assess the accuracy of various land cover products spanning the years from 2013 to 2022. We have added additional description in revised text (Lines 294-297), which reads:

*The land cover samples selected remained stable encompassing the years from 2013 to 2022 for all the other 4 land cover products, thus making them comparable to our TP_LC10-2022 map. Therefore, we validated the aggregation of samples into 8 categories and assessed the performance of TP_LC10-2022 and the 4 other land cover products in the TP region, as depicted in Table 5.*

**10. Are there plans to update the product annually or any other future research plans?**

**Reply 10:**

Yes, we have been producing land cover maps of the TP using Sentinel-2 imagery over the past few years, and we are updating the available samples in 2023, which will be available online soon. Please keep an eye on our data portal, where we are currently hosting *TP_LC10-2022 product*: *https://doi.org/10.5281/zenodo.8214981*

**11. Table 1, VV and VH are backscatter coefficients, not reflectivities. And it needs to be clarified with the direction of transmission and reception.**

**Reply 11:**

Thank you for the correction. We have revised the mistake and clarified the direction and transmission in the revision. To be clearer, we also added the orbit parameter of the Sentinel-1 satellite in Table 2.

**12. Typo in table 2, "DNSI" -> "NDSI".**

**Reply 12:**

Thank you for the correction, we have updated the table in our revised manuscript.

---

## Author Comment (AC3)

**A 10 m resolution land cover map of the Tibetan Plateau with detailed vegetation types**

Community Referee #1

**General comments:**

**This study proposes a 10 m resolution land cover map of the Tibetan Plateau with detailed vegetation types. The experimental design is thorough. However, I have some concerns.**

**Response:** We are grateful for your kind acknowledgment of the experimental design of our study and thank you for providing insightful comments and detailed suggestions. Following your constructive feedback, we have revised the text to strengthen the clarity and accuracy of this manuscript.

**Specific comments:**

1. **Line 89: Precipitation data is 0.05degree resolution, can the resampled 10m data maintain the quality?**

**Reply 1:**

The quality of resampled climate data is sufficient for the classification. Climate data contributes to the classification process mainly at the regional scale, for example, southeast TP is more humid than the middle and east parts of TP. At the local scale, satellite imagery and topography data play a dominant role in the classification process, while the contribution from climate data to classification is minimal. Besides, our product shows consistent finer spatial patterns at 10m resolution. For clarification, we have elaborated on this issue in the revised manuscript (Lines 255-258), which reads:

*The TP exhibits significant variations in annual rainfall and land surface temperature across its diverse regions, resulting in distinct hot and cold spots (Rao et al., 2019; Wu et al., 2019). Leveraging climate data can thus prove beneficial in categorizing alpine meadows in the southeastern TP and alpine grasslands in the northwestern TP at regional climatic scales, given their high sensitivity to changes in annual precipitation and land surface temperature (Su et al., 2020; Y. Wang et al., 2021).*

**Reference:**

Rao, Y., Liang, S., Wang, D., Yu, Y., Song, Z., Zhou, Y., Shen, M., & Xu, B. (2019). Estimating daily average surface air temperature using satellite land surface temperature and top-of-atmosphere radiation products over the Tibetan Plateau. Remote Sensing of Environment, 234, 111462. https://doi.org/10.1016/j.rse.2019.111462

Su, Y., Guo, Q., Hu, T., Guan, H., Jin, S., An, S., Chen, X., Guo, K., Hao, Z., Hu, Y., Huang, Y., Jiang, M., Li, J., Li, Z., Li, X., Li, X., Liang, C., Liu, R., Liu, Q., … Ma, K. (2020). An updated Vegetation Map of China (1:1000000). Science Bulletin, 65(13), 1125–1136. https://doi.org/10.1016/j.scib.2020.04.004

Wang, Y., Xiao, J., Ma, Y., Luo, Y., Hu, Z., Li, F., Li, Y., Gu, L., Li, Z., & Yuan, L. (2021). Carbon fluxes and environmental controls across different alpine grassland types on the Tibetan Plateau. Agricultural and Forest Meteorology, 311, 108694. https://doi.org/10.1016/j.agrformet.2021.108694

Wu, Y., Guo, L., Zheng, H., Zhang, B., & Li, M. (2019). Hydroclimate assessment of gridded precipitation products for the Tibetan Plateau. Science of The Total Environment, 660, 1555–1564. https://doi.org/10.1016/j.scitotenv.2019.01.119

**2.    Line 90: What is the spatial resolution of temperature data?**

**Reply 2:**

The spatial resolution of temperature data is 0.1degree. We also added this information to our revised manuscript (Line 103).

**3.    Line 116-117: Why combine bare land and impervious area? Because other land cover products usually separate these two classes.**

**Reply 3:**

Thanks for pointing this out. We have added an explanation and discussed this issue in our revised manuscript (Lines 132-136 and 266-268).

**Line 132-136:**

*In this study, we did not specifically select samples of built-up areas and instead categorized bare land together with built-up areas for two primary reasons. Firstly, built-up areas account for only 0.092% of the total area in ESA WorldCover2021, highlighting their relatively small extent compared to other land cover types (Zanaga et al., 2022). Secondly, bare land in our product exhibits spectral characteristics similar to those of built-up areas, resulting in the classification of most built-up areas as bare land (H. Li et al., 2017).*

**Lines 266-270:**

*In addition , the spectral variations within urban areas have also resulted in substantial uncertainties. Our approach of categorizing built-up areas and bare land may lead to misclassification of urban pixels. To minimize the uncertainties in urban areas on our final map, we applied the ESRI land cover map in 2022 to mask off urban pixels (Karra et al., 2021).*

**Reference:**

Karra, K., Kontgis, C., Statman-Weil, Z., Mazzariello, J. C., Mathis, M., & Brumby, S. P.

(2021). Global land use / land cover with Sentinel 2 and deep learning. 2021 IEEE International Geoscience and Remote Sensing Symposium IGARSS, 4704–4707. https://doi.org/10.1109/IGARSS47720.2021.9553499

Li, H., Wang, C., Zhong, C., Su, A., Xiong, C., Wang, J., & Liu, J. (2017). Mapping Urban Bare Land Automatically from Landsat Imagery with a Simple Index. Remote Sensing, 9(3), 249. https://doi.org/10.3390/rs9030249

Zanaga, D., Van De Kerchove, R., Daems, D., De Keersmaecker, W., Brockmann, C., Kirches, G., Wevers, J., Cartus, O., Santoro, M., Fritz, S., Lesiv, M., Herold, M., Tsendbazar, N.-E., Xu, P., Ramoino, F., and Arino, O.: ESA WorldCover 10 m 2021 v200, Zenodo [data set], https://doi.org/10.5281/zenodo.5571936, 2022

**4. Line 165: As the vegetation will be affected by seasons, have you considered getting the median composites of Sentinel data for each season, and then combining all seasons as the input?**

**Reply 4:**

We considered combining the four seasons as the input. However, generating seasonal composites requires acquiring high-quality Sentinel-2 time series images, which is challenging in the TP. For instance, lowering the threshold of cloud filtering results in the reduction of image pixels available for analysis, particularly in the southeastern TP, where has heavy cloud contamination (Tang et al., 2022). Conversely, raising this threshold to a higher level compromises the quality control of Sentinel-2 images while maintaining image integrity. We have included a quantitative discussion on this matter in the revised manuscript (Lines 273-278), which reads:

[Figure]

*Figure A3. Number of available observations for the Sentinel-2 optical data in the Tibetan Plateau during summer in 2022 (June 1, 2022, to August 31, 2022) with cloud cover <10%.*

*For example, during the summer of 2022 (June-August), when setting the "CLOUDY_PIXEL_PERCENTAGE" parameter to 10%, 20%, 30%, and 40%, and applying QA band masking, we lost 13.59%, 5.81%, 2.44%, and 1.32% of the Sentinel-2 image area in the TP. The removed pixels are concentrated mainly in the*

*cloudy southeastern TP (only shown for 10% threshold in Fig. A3) (Tang et al., 2022). This constraint can preclude the attainment of desired outcomes in regions where cloud-free image availability is low (Chu et al., 2021; Coluzzi et al., 2018).*

**Reference:**

Chu, D., Shen, H., Guan, X., Chen, J. M., Li, X., Li, J., & Zhang, L. (2021). Long time-series NDVI reconstruction in cloud-prone regions via spatio-temporal tensor completion. Remote Sensing of Environment, 264, 112632. https://doi.org/10.1016/j.rse.2021.112632

Coluzzi, R., Imbrenda, V., Lanfredi, M., & Simoniello, T. (2018). A first assessment of the Sentinel-2 Level 1-C cloud mask product to support informed surface analyses. Remote Sensing of Environment, 217, 426–443.

Tang, J., Guo, X., Chang, Y., Lu, G., & Qi, P. (2022). Long-term variations of clouds and precipitation on the Tibetan Plateau and its subregions, and the associated mechanisms. International Journal of Climatology, 42(16), 9003–9022. https://doi.org/10.1002/joc.7792

5. **Line 180: This study is based on pixel-based machine learning classification models. The pixel-based approach tends to produce classification with a salt-pepper effect, did you do any post-classification to remove the noise?**

**Reply 5:**

Thank you for raising this concern. We did not perform any post-classification noise removal methods. The TP exhibits highly heterogeneous vegetation landscapes. Applying image smoothing techniques to eliminate noise may not accurately represent the diverse distribution of vegetation types and could lead to a loss of detailed edge information.

6. **Line 185: Why not use the major voting results of all models as the final results?**

**Reply 6:**

We selected the model with best performance to ensure consistency in our final map across the entire TP, while different model exhibit variations in the classification performance for different land cover types, choosing appropriate weights for each model might be challenging.

7. **Line 245: Have you considered comparing the area in each land cover between your classification and other land cover products?**

**Reply 7:**

Thank you for this valuable suggestion. We have included a new table in the revised manuscript to address this aspect.

*Table A4 presents the statistical results of 5 land cover products in the TP, highlighting significant discrepancies among them.*

*Table A4. Area statistical results for land cover products in the Tibetan Plateau.*

| Land cover type | TP_LC10-2022 | | FROM_GLC30-2015 | | GLC_FCS30-2020 | | WorldCover2021 | | FROM_GLC10-2017 | |
|---|---|---|---|---|---|---|---|---|---|---|
| | Area | Proportion | Area | Proportion | Area | Proportion | Area | Proportion | Area | Proportion |
| BL | 58.75 | 19.05% | 147.67 | 47.89% | 45.71 | 14.82% | 134.75 | 43.70% | 156.45 | 50.74% |
| AG | 50.83 | 16.48% | 96.75 | 31.38% | 189.44 | 61.44% | 108.44 | 35.17% | 89.35 | 28.98% |
| AM | 73.25 | 23.76% | | | | | | | | |
| ENF | 11.44 | 3.71% | 27.91 | 9.05% | 31.52 | 10.22% | | | | |
| DNF | 2.26 | 0.73% | 0.02 | 0.01% | 0.37 | 0.12% | | | | |
| EBF | 4.53 | 1.47% | 2.94 | 0.95% | 3.45 | 1.12% | 28.49 | 9.24% | 29.46 | 9.55% |
| DBF | 6.80 | 2.20% | 1.69 | 0.55% | 4.31 | 1.40% | | | | |
| MF | 1.46 | 0.47% | 4.10 | 1.33% | 0.00 | 0.00% | | | | |
| ES | 3.28 | 1.06% | 1.70 | 0.55% | 0.22 | 0.07% | 0.37 | 0.12% | 1.59 | 0.51% |
| DS | 11.02 | 3.57% | 0.41 | 0.13% | 4.13 | 1.34% | | | | |
| WB | 6.43 | 2.09% | 12.38 | 4.02% | 6.05 | 1.96% | 6.86 | 2.22% | 10.06 | 3.26% |
| WL | 6.84 | 2.22% | 0.19 | 0.06% | 0.55 | 0.18% | 0.37 | 0.12% | 2.06 | 0.67% |
| CV | 5.14 | 1.67% | 2.04 | 0.66% | 2.81 | 0.91% | 1.35 | 0.44% | 3.02 | 0.98% |
| PIS | 23.18 | 7.52% | 10.46 | 3.39% | 19.08 | 6.19% | 12.95 | 4.20% | 16.36 | 5.30% |
| AS / | | | | | | | | | | |
| / Tundra | 43.15 | 13.99% | 0.05 | 0.01% | 0.00 | 0.00% | 14.77 | 4.79% | | |
| Lichen / Moss | | | | | | | | | 0.00 | 0.00% |
| Total | 308.34 | 100.00% | 308.31 | 99.99% | 307.66 | 99.78% | 308.34 | 100.00% | 308.34 | 100.00% |

BL: bare land; AG: alpine grassland; AM: alpine meadow; ENF: evergreen needle-leaved forest; DNF: deciduous needle-leaved forest; EBF: evergreen broadleaved forest; DBF: deciduous broadleaved forest; MF: mixed forest; ES: evergreen shrubland; DS: deciduous shrubland; WB: water body; WL: wetland; CV: cultivated vegetation; PIS: permanent ice and snow; AS: alpine scree

• The unit of area is ten thousand square kilometers, and the unit of proportion is percent.

*• Please refer to Table A3 for the merging rules of land cover for FROM_GLC30-2015 and GLC_FCS30-2020.*
*• The 'cloud' class in the FROM_GLC30-2015 and 'shrubland' class in the GLC_FCS30-2020 product have been omitted from the table due to their small area.*
*• All built-up pixels are merged with bare land.*

Also, we have strengthened the discussion using this table in the revised manuscript (Lines 251-254 and 322-328)

**Lines 251-254:**

*GLC_FCS30-2020 exhibits the highest consistency with TP_LC10-2022 regarding bare land (Table A4 and Fig. 5), but it classified more areas as grasslands while failing to differentiate between grasslands and meadows. According to Fig. 5b, ESA WorldCover2021, FROM_GLC10-2017, and FROM_GLC30-2015 products overestimate the area of bare land in the TP, similar to the issues observed in FROM_GLC-agg and ESA CCI land cover products (Liu et al., 2021; L. Yu et al., 2014). This may be due to the misclassification of alpine grassland as bare land because these products captured less spectral information during the growing season of alpine grasslands. GLC_FCS30-2020 exhibits the highest consistency with TP_LC10-2022 regarding bare land (Table A4 and Fig. 5) and it classified more grasslands while failed to differentiate between grasslands and meadows. Additionally, GLC_FCS30-2020 assigns 61.44% of the total TP area as grassland, indicating an overestimation of grassland extent (Table A4).*

**Lines 320-326:**

*Alpine forests play a crucial role in carbon storage and sequestration, thereby enhancing ecosystem services in the TP (Lin et al., 2023; Z. Wang et al., 2022; H. Zhao et al., 2023). Our study revealed that TP_LC10-2022 identified the smallest forested area (8.60%), while GLC_FCS30-2020 and FROM_GLC30-2015 classified the largest and second-largest areas of alpine forest, respectively (12.86% and 11.89%) (Table A4). Conversely, the area of shrubland exhibits nearly the opposite trend (Table A4). Confusion also arises between alpine grassland and bare land, potentially leading to variations in carbon storage estimation within each vegetation type. These discrepancies could impact efforts related to forest resource protection and grassland management for animal husbandry (J. Li et al., 2020; C. Yu et al., 2022).*

**Reference:**

Cheng, Y. (2021). Climate response to introduction of the ESA CCI land cover data to the NCAR CESM. Climate Dynamics, 56(11–12), 4109–4127. https://doi.org/10.1007/s00382-021-05690-3

Li, J., Gong, J., Guldmann, J.-M., Li, S., & Zhu, J. (2020). Carbon Dynamics in the

Northeastern Qinghai–Tibetan Plateau from 1990 to 2030 Using Landsat Land Use/Cover Change Data. Remote Sensing, 12(3), 528. https://doi.org/10.3390/rs12030528

Lin, Y., Xiao, J.-T., Kou, Y.-P., Zu, J.-X., Yu, X.-R., & Li, Y.-Y. (2023). Aboveground carbon sequestration rate in alpine forests on the eastern Tibetan Plateau: Impacts of future forest management options. Journal of Plant Ecology, 16(3), rtad001. https://doi.org/10.1093/jpe/rtad001

Wang, Z., Song, W., & Yin, L. (2022). Responses in ecosystem services to projected land cover changes on the Tibetan Plateau. Ecological Indicators, 142, 109228. https://doi.org/10.1016/j.ecolind.2022.109228

Yu, C., Xu, L., Li, M., & He, N. (2022). Phosphorus storage and allocation in vegetation on the Tibetan Plateau. Ecological Indicators, 145, 109636. https://doi.org/10.1016/j.ecolind.2022.109636

Yu, L., Wang, J., Li, X., Li, C., Zhao, Y., & Gong, P. (2014). A multi-resolution global land cover dataset through multisource data aggregation. Science China Earth Sciences, 57(10), 2317–2329. https://doi.org/10.1007/s11430-014-4919-z

Zhao, H., Guo, B., & Wang, G. (2023). Spatial–Temporal Changes and Prediction of Carbon Storage in the Tibetan Plateau Based on PLUS-InVEST Model. Forests, 14(7), 1352. https://doi.org/10.3390/f14071352

**8. Authors need to elaborate on the discussion section using more references and describe the implications of your product for the sustainable use of available resources in practice, for policy, and research.**

**Reply 8:**

Thank you for your constructive advice. We have elaborated our discussion in terms of its implications for policy, research, and the sustainable use of available resources in practice (Lines 312-340).

**1. For sustainable use of available resources in practice:**

*Lakes and glaciers are the sentinels of global climate change and constitute the foundation of the TP as a crucial water source for surrounding regions (G. Zhang et al., 2017; G. Zhang & Duan, 2021). Precisely extracting the boundaries of lakes and glaciers is imperative for quantitatively monitoring lake expansion and glacier melting, as well as understanding the dynamic relationship between them and precipitation (Tong et al., 2016; J. Zhang et al., 2021; R. Zhao et al., 2022). Our land cover data, samples, and mapping methodology can serve as a baseline support for these endeavors (Korzeniowska & Korup, 2017; Yan et al., 2020), which facilitates the effective utilization of available water resources and promotes the sustainable development of the economy and society in the Greater Tibetan Plateau area and downstream regions of rivers originating from the TP (Ding et al., 2019).*

**2. For policy making:**

*Alpine forests play a crucial role in carbon storage and sequestration, thereby enhancing ecosystem services in the TP (Lin et al., 2023; Z. Wang et al., 2022; H. Zhao et al., 2023). Our study revealed that TP_LC10-2022 identified the smallest forested area (8.60%), while GLC_FCS30-2020 and FROM_GLC30-2015 classified the largest and second-largest areas of alpine forest, respectively (12.86% and 11.89%) (Table A4). Conversely, the area of shrubland exhibits nearly the opposite trend (Table A4). Confusion also arises between alpine grassland and bare land, potentially leading to variations in carbon storage estimation within each vegetation type. These discrepancies could impact efforts related to forest resource protection and grassland management for animal husbandry (J. Li et al., 2020; C. Yu et al., 2022).*

**3. For further research:**

*Alpine screes are extensively distributed across the TP, yet they are frequently disregarded from other products. Our product presents the initial description of alpine scree vegetation locations, which will contribute to environmental monitoring and biodiversity research in the periglacial zone of the TP (X.-H. Li et al., 2014). Shrublands play a vital role as carbon sinks in ecosystems and hold substantial implications for biomass estimation and global carbon cycling (Ma et al., 2021; Nie et al., 2018). TP_LC10-2022 accurately predicts the spatial distribution of shrublands, which holds considerable importance in forecasting the impact of future changes in the biomass and carbon cycle on global-scale ecosystems (Chang et al., 2022).*

*High-resolution and accurate land cover data encompassing diverse vegetation types are crucial for monitoring large-scale alpine vegetation dynamics (F. Wang et al., 2023; Z. Wang et al., 2020, 2022). For instance, relying on land cover maps such as ESA WorldCover as the foundation to examine tree lines and vegetation lines in the TP may lead to the underestimation of tree lines due to misclassifications of grasslands and shrublands (Fig. 7) (Zou et al., 2023). Additionally, the vegetation line may also be underestimated because of the absence of alpine scree (Fig. 7). In our future work, we aim to leverage the Sentinel-2, Sentinel-1, and other multisource data to annually generate TP_LC10 products. This approach will facilitate alpine vegetation monitoring and change detection, thereby enriching our comprehension of the dynamic TP amidst intensifying global climate change (Y. Wang et al., 2022).*

**References:**

Chang, Q., Zwieback, S., DeVries, B., & Berg, A. (2022). Application of L-band SAR for mapping tundra shrub biomass, leaf area index, and rainfall interception. Remote Sensing of Environment, 268, 112747. https://doi.org/10.1016/j.rse.2021.112747

Ding, X., Zhang, Z., Wu, F., & Xu, X. (2019). Study on the Evolution of Water Resource Utilization Efficiency in Tibet Autonomous Region and Four Provinces in Tibetan

Areas under Double Control Action. Sustainability, 11(12), 3396. https://doi.org/10.3390/su11123396

Korzeniowska, K., & Korup, O. (2017). Object-Based Detection of Lakes Prone to Seasonal Ice Cover on the Tibetan Plateau. Remote Sensing, 9(4), 339. https://doi.org/10.3390/rs9040339

Li, J., Gong, J., Guldmann, J.-M., Li, S., & Zhu, J. (2020). Carbon Dynamics in the Northeastern Qinghai–Tibetan Plateau from 1990 to 2030 Using Landsat Land Use/Cover Change Data. Remote Sensing, 12(3), 528. https://doi.org/10.3390/rs12030528

Li, X.-H., Zhu, X.-X., Niu, Y., & Sun, H. (2014). Phylogenetic clustering and overdispersion for alpine plants along elevational gradient in the Hengduan Mountains Region, southwest China: Phylogenetic structure along elevational gradient. Journal of Systematics and Evolution, 52(3), 280–288. https://doi.org/10.1111/jse.12027

Lin, Y., Xiao, J.-T., Kou, Y.-P., Zu, J.-X., Yu, X.-R., & Li, Y.-Y. (2023). Aboveground carbon sequestration rate in alpine forests on the eastern Tibetan Plateau: Impacts of future forest management options. Journal of Plant Ecology, 16(3), rtad001. https://doi.org/10.1093/jpe/rtad001

Ma, H., Mo, L., Crowther, T. W., Maynard, D. S., van den Hoogen, J., Stocker, B. D., Terrer, C., & Zohner, C. M. (2021). The global distribution and environmental drivers of aboveground versus belowground plant biomass. Nature Ecology & Evolution, 5(8), 1110–1122. https://doi.org/10.1038/s41559-021-01485-1

Nie, X., Yang, L., Xiong, F., Li, C., Fan, L., & Zhou, G. (2018). Aboveground biomass of the alpine shrub ecosystems in Three-River Source Region of the Tibetan Plateau. Journal of Mountain Science, 15(2), 357–363. https://doi.org/10.1007/s11629-016-4337-0

Tong, K., Su, F., & Xu, B. (2016). Quantifying the contribution of glacier meltwater in the expansion of the largest lake in Tibet. Journal of Geophysical Research: Atmospheres, 121(19). https://doi.org/10.1002/2016JD025424

Wang, F., Ma, Y., Darvishzadeh, R., & Han, C. (2023). Annual and Seasonal Trends of Vegetation Responses and Feedback to Temperature on the Tibetan Plateau since the 1980s. Remote Sensing, 15(9), 2475. https://doi.org/10.3390/rs15092475

Wang, Y., Li, D., Ren, P., Ram Sigdel, S., & Camarero, J. J. (2022). Heterogeneous Responses of Alpine Treelines to Climate Warming across the Tibetan Plateau. Forests, 13(5), 788. https://doi.org/10.3390/f13050788

Wang, Z., Song, W., & Yin, L. (2022). Responses in ecosystem services to projected land cover changes on the Tibetan Plateau. Ecological Indicators, 142, 109228. https://doi.org/10.1016/j.ecolind.2022.109228

Wang, Z., Wu, J., Niu, B., He, Y., Zu, J., Li, M., & Zhang, X. (2020). Vegetation Expansion on the Tibetan Plateau and Its Relationship with Climate Change. Remote Sensing, 12(24), 4150. https://doi.org/10.3390/rs12244150

Yan, D., Huang, C., Ma, N., & Zhang, Y. (2020). Improved Landsat-Based Water and Snow Indices for Extracting Lake and Snow Cover/Glacier in the Tibetan Plateau. Water, 12(5), 1339. https://doi.org/10.3390/w12051339

Yu, C., Xu, L., Li, M., & He, N. (2022). Phosphorus storage and allocation in vegetation on the Tibetan Plateau. Ecological Indicators, 145, 109636.

https://doi.org/10.1016/j.ecolind.2022.109636

Zhang, G., & Duan, S. (2021). Lakes as sentinels of climate change on the Tibetan Plateau. All Earth, 33(1), 161–165. https://doi.org/10.1080/27669645.2021.2015870

Zhang, G., Yao, T., Piao, S., Bolch, T., Xie, H., Chen, D., Gao, Y., O'Reilly, C. M., Shum, C. K., Yang, K., Yi, S., Lei, Y., Wang, W., He, Y., Shang, K., Yang, X., & Zhang, H. (2017). Extensive and drastically different alpine lake changes on Asia's high plateaus during the past four decades. Geophysical Research Letters, 44(1), 252–260. https://doi.org/10.1002/2016GL072033

Zhang, J., Hu, Q., Li, Y., Li, H., & Li, J. (2021). Area, lake-level and volume variations of typical lakes on the Tibetan Plateau and their response to climate change, 1972–2019. Geo-Spatial Information Science, 24(3), 458–473. https://doi.org/10.1080/10095020.2021.1940318

Zhao, H., Guo, B., & Wang, G. (2023). Spatial–Temporal Changes and Prediction of Carbon Storage in the Tibetan Plateau Based on PLUS-InVEST Model. Forests, 14(7), 1352. https://doi.org/10.3390/f14071352

Zhao, R., Fu, P., Zhou, Y., Xiao, X., Grebby, S., Zhang, G., & Dong, J. (2022). Annual 30-m big Lake Maps of the Tibetan Plateau in 1991–2018. Scientific Data, 9(1), 164. https://doi.org/10.1038/s41597-022-01275-9

Zou, L., Tian, F., Liang, T., Eklundh, L., Tong, X., Tagesson, T., Dou, Y., He, T., Liang, S., & Fensholt, R. (2023). Assessing the upper elevational limits of vegetation growth in global high-mountains. Remote Sensing of Environment, 286, 113423. https://doi.org/10.1016/j.rse.2022.113423

---

## Author Response (AR2)

**A 10 m resolution land cover map of the Tibetan Plateau with detailed vegetation types**

Anonymous Referee #2

**General comments:**

**The authors have presented revisions to justify Data description and Methods, and incorporated most suggestions in doing so. The manuscript is improved. I am pleased to see the careful revisions made by the authors. However, the manuscript still needs few minor content revisions.**

**Response:** We thank you for providing detailed minor revision comments. We have incorporated the comments to improve the manuscript.

**Specific comments:**

1. **First, I do not agree with the authors' response regarding the use of median composites for mapping. Median composites are greatly influenced by the number of available images, so in many places, it is actually not possible to achieve the effect shown in Figure A4. Additionally, the multi-source information mentioned by the authors mainly emphasizes DEM information, which, although helpful for distinguishing vegetation, is still limited. For differentiating various types of vegetation, multi-temporal information is definitely the most helpful. I suggest that the authors mention this issue in the Limitations section.**

**Reply 1:**

We agree with you that the median composites are affected by the number of available images. Therefore, we have added an explanation in the revised manuscript (Lines 286-289), which reads:

*Median composites are affected by the number of available images, we thus ensured a minimum of three high-quality observations across the entire TP while preprocessing the annual Sentinel-2 images. The composites from ≥ three Sentinel images make it possible to achieve the seamless effect shown in Figure A4 in various locations over large areas of the TP.*

Additionally, information derived from DEM is limited to differentiating vegetation types in certain regions such as flat or urban areas. We have included this limitation regarding the utilization of topography dataset in the revised manuscript (Lines 295-298), which reads:

*Conversely, in flat areas where vegetation distribution is minimally influenced by topography, or in urban areas where vegetation distribution is affected by*

*anthropogenic activity, topographic information may exhibit limitations in land cover classification (Zeng et al., 2019).*

We agree that multi-temporal information is more helpful than single median composites, although median aggregation is computationally efficient for large-scale research and achieved good quality in this study. In fact, we emphasized this view in every version of our manuscript (Lines 272-275), which reads:

*Although we employed the Sentinel-2 median composition method in this study, we acknowledge the potential enhancement that time-series analysis could bring to our research. In comparison to median composition, time-series analysis has the potential to more comprehensively capture phenological information of vegetation, thereby yielding more accurate land cover classification results (Xie et al., 2019; Nguyen et al., 2020).*

**Reference:**

Nguyen, L. H., Joshi, D. R., Clay, D. E., and Henebry, G. M.: Characterizing land cover/land use from multiple years of Landsat and MODIS time series: A novel approach using land surface phenology modeling and random forest classifier, Remote Sens. Environ., 238, 111 017, https://doi.org/10.1016/j.rse.2018.12.016, 2020

Xie, S., Liu, L., Zhang, X., Yang, J., Chen, X., and Gao, Y.: Automatic land-cover mapping using landsat time-series data based on google earth engine, Remote Sens., 11, 3023, https://doi.org/10.3390/rs11243023, 2019

Zeng, T., Wang, L., Zhang, Z., Wen, Q., Wang, X., & Yu, L. (2019). An Integrated Land Cover Mapping Method Suitable for Low-Accuracy Areas in Global Land Cover Maps. Remote Sensing, 11(15), 1777. https://doi.org/10.3390/rs11151777

2. **Moreover, it is also suggested that the authors include in the Limitations section the potential impact on classification results due to the direct use of coarse-resolution CHIRPS and ERA5 data.**

**Reply 2:**

Thank you for this suggestion. The direct use of coarse-resolution data may lead to potential loss of spatial information. However, we have found that its incorporation can enhance the accuracy of the results, and we did not observe any negative impact on the fine-scaled spatial patterns of our classification results. We have included an explanation in the revised manuscript (Lines 259-261), which reads:

*Our study also found incorporating resampled coarse-resolution climate data can help improve the classification accuracy of finer resolution data (Jia et al., 2014). However, it may cause potential loss of spatial information (Xu et al., 2020), which has not been observed in the TP_LC10-2022 dataset.*

**Reference:**

Jia, K., Liang, S., Zhang, N., Wei, X., Gu, X., Zhao, X., Yao, Y., & Xie, X. (2014). Land cover

classification of finer resolution remote sensing data integrating temporal features from time series coarser resolution data. ISPRS Journal of Photogrammetry and Remote Sensing, 93, 49–55. https://doi.org/10.1016/j.isprsjprs.2014.04.004

Xu, Y., Yu, L., Peng, D., Zhao, J., Cheng, Y., Liu, X., Li, W., Meng, R., Xu, X., & Gong, P. (2020). Annual 30-m land use/land cover maps of China for 1980–2015 from the integration of AVHRR, MODIS and Landsat data using the BFAST algorithm. Science China Earth Sciences, 63(9), 1390–1407. https://doi.org/10.1007/s11430-019-9606-4